# The Dormant Neuron Phenomenon in Multi-Agent Reinforcement Learning Value Factorization

**Haoyuan Qin**$^{ab*}$, **Chennan Ma**$^{ab*}$, **Mian Deng**$^{ab}$, **Zhengzhu Liu**$^{ab}$,
**Songzhu Mei**$^{c}$, **Xinwang Liu**$^{c}$, **Cheng Wang**$^{ab}$, **Siqi Shen**$^{ab\dagger}$

$^a$Fujian Key Laboratory of Sensing and Computing for Smart Cities,
School of Informatics, Xiamen University (XMU), China
$^b$Key Laboratory of Multimedia Trusted Perception and Efficient Computing, XMU, China
$^c$School of Computer, National University of Defense Technology, China
{haoyuanqin,chennanma}@stu.xmu.edu.cn, {cwang,siqishen}@xmu.edu.cn,
{sz.mei,xinwangliu}@nudt.edu.cn

## Abstract

In this work, we study the dormant neuron phenomenon in multi-agent reinforcement learning value factorization, where the mixing network suffers from reduced network expressivity caused by an increasing number of inactive neurons. We demonstrate the presence of the dormant neuron phenomenon across multiple environments and algorithms, and show that this phenomenon negatively affects the learning process. We show that dormant neurons correlates with the existence of over-active neurons, which have large activation scores. To address the dormant neuron issue, we propose ReBorn, a simple but effective method that transfers the weights from over-active neurons to dormant neurons. We theoretically show that this method can ensure the learned action preferences are not forgotten after the weight-transferring procedure, which increases learning effectiveness. Our extensive experiments reveal that ReBorn achieves promising results across various environments and improves the performance of multiple popular value factorization approaches. The source code of ReBorn is available in https://github.com/xmu-rl-3dv/ReBorn.

## 1 Introduction

In cooperative Multi-Agent Reinforcement Learning (MARL) [1], a group of agents whose value function is approximated using deep neural networks must cooperate to achieve a common goal. Deep neural network is the key driving force that scales MARL for complex decision-making tasks [2]. Recently, researchers have discovered the scaling laws that deep neural network models can increase their capacity by enlarging the size of the model and the dataset. However, single-agent reinforcement learning does not obey the scaling law and suffers from network expressivity issues [3, 4].

To alleviate the network expressivity issues in single-agent RL, researchers have proposed parameter perturbing methods. Igl et al. [5] periodically *resets* some layers of the value network. ReSet [6] resets the last few layers of the neural network while maintaining experience in the replay buffer. ReDo [4] periodically re-initializes the input weights of some neurons and zero out the neuron's output weights. Albeit these methods can improve the performance of single-agent reinforcement learning, it is unclear whether they work for MARL.

---

$^*$Equal contribution
$^\dagger$Corresponding author

38th Conference on Neural Information Processing Systems (NeurIPS 2024).

Compared to single-agent RL, MARL is more challenging, including issues such as partial-observability [7] and the non-stationary of other learning agents' policies. The Centralized Training with Decentralized Execution (CTDE) paradigm [8] is widely adopted in this context. In CTDE, it is a common practice to use value factorization [2, 9], which factorizes a joint state-action value function $Q_{tot}$ into individual agent utilities $Q_i$. Each agent acts according to $Q_i$, which is approximated using a deep neural network, named the *agent network*. $Q_i$ are mixed through a neural network, the *mixing network*, to form joint value function $Q_{tot}$.

In this work, we explore the reasons behind the reduction in network expressivity issues in cooperative MARL. Specifically, we study *dormant neurons* [4], which remain inactive with low activation levels during learning. We demonstrate that the dormant neuron phenomenon, the number of dormant neurons increases during the training process, exists in multiple popular value-based MARL algorithms (QMIX [2], QPLEX [10], DMIX [11], and RMIX [12]) across various environments (e.g., SMAC [13], SMACv2 [14], predator prey [15]). We find that the proportion of dormant neurons increases with the number of agents, and that dormant neurons mainly exist in the mixing network. Moreover, we identify the existence of *over-active neurons*, whose activation score accounts for a significant portion of the activation scores for all the neurons.

Typical network parameter perturbing approaches used in single-agent RL (such as Reset [5, 6] and ReDo [4]) do not work efficiently in MARL. Parameter perturbing methods, which change the weights of neurons, may lead to forgetting of learned knowledge, especially in MARL with high cooperation demands. The cooperation knowledge should not be forgotten even after parameter perturbation. We formulate a memorization requirement that the learned cooperative action preferences remain unchanged after parameter perturbation as the *Knowledge Invariant (KI) principle*. We theoretically show that existing approaches [5, 6, 4] cannot guarantee adherence to the KI principle. Failing to satisfy the KI principle can lead to the violation of the Individual-Global-Max (IGM) principle, which is widely adopted in MARL.

We propose, *ReBorn*, a simple but effective method that transfers the weights from over-active neurons to dormant neurons. It periodically detects dormant and over-active neurons, and balances the weights among them. We theoretically show that ReBorn satisfies the KI principle for various value factorization approaches (e.g., QMIX and QPLEX), distributional value factorization approaches (i.e., DMIX and DDN [11]), and risk-sensitive value factorization approach (i.e., RMIX [12]). Through extensive experiments, we demonstrate that ReBorn can improve the performance of multiple MARL value factorization methods, and it performs better than multiple parameter perturbing methods by effectively remembering previously learned knowledge.

## 2 Background

### 2.1 Dec-POMDPs

We consider Decentralized Partially Observable Markov Decision Processes (Dec-POMDPs) [16] in modeling cooperative multi-agent reinforcement learning (MARL) scenarios. A Dec-POMDP can be defined by a tuple $G = \langle \mathcal{S}, \{\mathcal{U}_i\}_{i=1}^{N}, P, r, \{\mathcal{O}_i\}_{i=1}^{N}, \{\sigma_i\}_{i=1}^{N}, N, \gamma \rangle$, where $\mathcal{N}$ is the set of agents, $\mathcal{S}$ is the states set, and $\mathcal{U}_i$ is the action set for agent $i$. At time step $t$, each agent $i$ chooses an action $u_i^t$, forming a joint action $\boldsymbol{u}^t$, leading to a state transition $s^{t+1} \sim P(\cdot|s^t, \boldsymbol{u}^t)$ and a joint reward $r^t$. In consideration of partial observability, each agent can only make decisions based on its local observation $o_i^t \sim \sigma^i(\cdot|s^t) \in \mathcal{O}_i$. Each agent $i$ act according to its individual policy $\pi_i(u_i|\tau_i)$ based on its local action-observation history $\tau_i = (O_i \times U_i)^*$, forming a joint policy $\pi = <\pi_1, \ldots, \pi_N>$. The joint policy $\pi$ has a joint action-value function: $Q^\pi(s_t, \boldsymbol{u}_t) = \mathbb{E}_{s_{t+1:\infty}, \boldsymbol{u}_{t+1:\infty}}[R_t \mid s_t, \boldsymbol{u}_t]$, where $R_t = \sum_{i=0}^{\infty} \gamma^i r_{t+i}$ is the discounted return, $\gamma$ is the discounting factor.

### 2.2 Value Function Factorization

In value factorization methods [17, 2, 9, 10, 18], per-agent utilities $Q_i$ is approximated using the *agent network*, and they are mixed through the *mixer network* to form the joint state-action value function $Q_{tot}$. For value factorization, the Individual-Global-Max (IGM) principle [9] is a critical criterion that ensure the consistency between local and joint optimal action selections. It is defined as follows:

**Definition 1** (IGM [9]). *For a joint state-action value function $Q_{jt} : \mathcal{T}^N \times \mathcal{U}^N \mapsto \mathbb{R}$, where $\tau \in \mathcal{T}^N$ is a joint action-observation history and $\mathbf{u}$ is the joint action, if there exists individual state-action functions $[Q_i : \mathcal{T}_i \times \mathcal{U}_i \mapsto \mathbb{R}]_{i=1}^N$, such that the following conditions are satisfied*

$$\arg\max_{\mathbf{u}} Q_{jt}(\boldsymbol{\tau}, \boldsymbol{u}) = (\arg\max_{u_1} Q_1(\tau_1, u_1), \ \ldots, \ \arg\max_{u_n} Q_N(\tau_N, u_N)), \tag{1}$$

*then, we can state that $[Q_i]_{i=1}^N$ satisfy IGM for $Q_{jt}$ under $\tau$, or $Q_{jt}(\boldsymbol{\tau}, \boldsymbol{u})$ is factorized by $[Q_i(\tau_i, u_i)]_{i=1}^N$.*

### 2.3 The Dormant Neuron Phenomenon

**Definition 2** ($\alpha-$dormant neuron [4, 19]). *Consider a fully connected layer $\ell$ within a neural network, where $H^\ell$ denotes the total number of neurons in this layer. For an input distribution $D$, let $h_i^\ell(x)$ represent the activation of neuron $i$ in layer $\ell$ under input $x \in D$. The normalized activation score of neuron $i$ is defined as follows:*

$$s_i^\ell = \frac{\mathbb{E}_{x \in \mathcal{D}} |h_i^\ell(x)|}{\frac{1}{H^\ell} \sum_{k=1}^{H^\ell} \mathbb{E}_{x \in \mathcal{D}} |h_k^\ell(x)|} \tag{2}$$

*Then a neuron $i$ in layer $\ell$ can be defined as $\alpha$-dormant if its score $s_i^\ell \leq \alpha$. (i.e., 0.1)*

**Definition 3** ($\alpha-$dormant ratio [4]). *The $\alpha$-**dormant ratio** of a neural network $\phi$ can be defined as follows:*

$$\beta_\alpha = \sum_{\ell \in \phi} N_\alpha^\ell / \sum_{\ell \in \phi} H^\ell \tag{3}$$

$N_\alpha^\ell$ *is the count of neurons that are $\alpha$-dormant in layer $\ell$, $H^\ell$ is number of neurons in layer $\ell$.*

The **dormant neuron phenomenon** refers to the steady increase in the dormant ratio of the neural network throughout training.

## 3 Related Work

### 3.1 Value Factorization

Value factorization approaches [20] are widely adopted in MARL. These methods construct the joint state-action value function $Q_{tot}$ based on individual utility $Q_i$. VDN [17] models the joint value function as the sum of individual utility function, while QMIX [2] models the monotonic increasing relationship among $Q_{tot}$ and $Q_i$. Qatten [18] models the relationship through using the attention mechanism. QPLEX [10] factorizes $Q_{tot}$ into a value function and an advantage function. QTRAN [9] and ResQ [21] decompose the value function into easy-to-factorized forms. For distributional MARL, DMIX [11] factorizes value function through mean-shape decomposition. A few work [12, 22] explore risk-sensitive value factorization. RMIX [12] models the monotonic increasing relationship among $Q_{tot}$ and the CVaR measure of each agent's distributional utility. RiskQ [22] ensures that the collection of greedy selection of risk-sensitive individual actions is equal to the greedy selection of risk-sensitive joint actions.

These methods focus on modeling the representation ability and functional relationships between the joint state-action value function and individual utilities. Our work, ReBorn, is orthogonal to these approaches, can be used to improve their overall performance by reducing dormant neurons.

### 3.2 RL neural network expressivity

In deep reinforcement learning, neural networks tend to lose their expressive power as training progresses [3]. Various studies explore the loss of expressiveness from different perspectives and propose corresponding methods to mitigate this issue.

Lyle et al. [23] show that the instability of the target can cause the network to lose expressive ability. ReSet [6] addresses early agent experience bias by periodically resetting the last layer of the neural network. The loss of expressive ability can also be attributed to over-fitting, a phenomenon analyzed in depth by Kirk et al. [24] and Zhang et al. [25] within reinforcement learning.

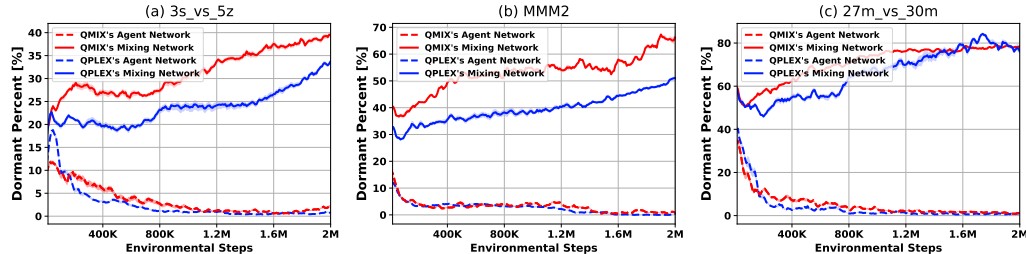

Figure 1: The existence of Dormant Neuron Phenomenon in Value Function Factorization Methods.

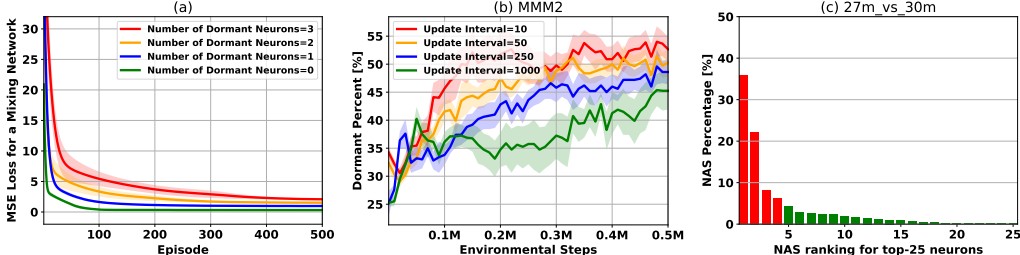

Figure 2: (a) The MSE Loss for fitting a simple Mixing Network increases with an increasing number of Dormant Neurons. It indicates that dormant neurons hurt mixing network expressivity. (b) The percentage of dormant neurons in QMIX mixing network with different target network update intervals. (c) The Normalized Activation Score (NAS) percentage ranking for top-25 over-active neurons in the QMIX mixing network.

To enhance generalization, researchers propose network randomization [26], convolution architectures [27], and soft data augmentation [28]. Researchers [29] find that the loss of plasticity is deeply connected to changes in the curvature of the loss landscape, and plasticity injection [30] is used to enhance the learning ability of neural networks for new data. D'Oro et al. [31] and Yang et al. [32] propose Reset Replay to improve the sample efficiency. ReDo [4] discovers that dormant neurons occur due to the instability of the target in reinforcement learning. DRM [19] finds that the dormant neuron phenomenon is related to agent exploration. When the dormancy ratio is high, the agent gradually cease exploration.

ReBorn is a parameter perturbing method for MARL. It can effectively reduce the number of dormant and over-active neurons. Moreover, it ensures that learned action knowledge is not forgotten after parameter perturbation.

## 4    The Dormant Neuron Phenomenon in MARL

**Dormant neurons mainly exist in the mixing network of MARL.** To verify the existence of the dormant neuron phenomenon in MARL, we analyze the number of dormant neurons during the training of QMIX [2] and QPLEX [10] across multiple tasks in SMAC [13]. The percentage of dormant neurons are illustrated in Figure 1, presented separately for the agent and the mixing networks. We discover that the dormant neuron phenomenon *primarily occurs in the mixing network* of MARL. The percentage of dormant neurons the mixing network is initially high and continues to increase, while the percentage of dormant neurons in agent networks is low. This observation is consistent across various algorithms and environments as it is depicted in Appendix D.4. The number of agents in the three tasks are 3, 10, and 27, respectively. As shown in Figure 1 (a) to (c), *with the increasing number of agents, the percentage of dormant neurons increases.*

**Dormant neurons hurt the expressive power of mixing networks.** In MARL value factorization methods, the mixing network plays a crucial role in integrating individual utilities into a joint value function. As shown in [9, 20, 21], the expressive power of the mixing network significantly impacts the performance of MARL value factorization methods. The expressive power of neural networks is related to both their depth [33, 34] and width [35, 36, 37]. We study expressive power of mixing networks from the perspective of dormant neurons, We use a mixing network with 2 Multi-layer Perceptron (MLP) layers, and fit it to a simple value function. This network consists of 4 neurons, and

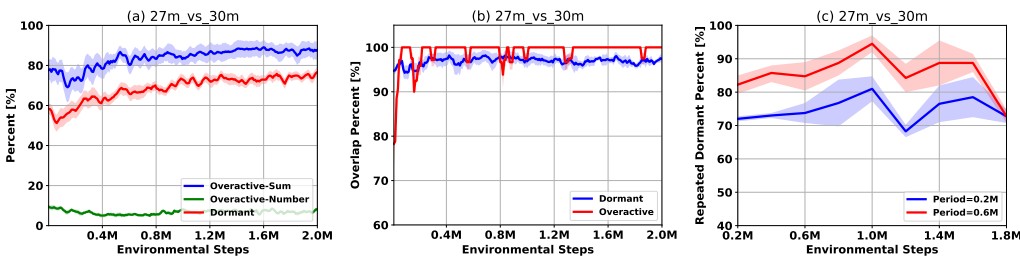

Figure 3: Over-active neurons in QMIX mixing networks: (a) The percentage contribution of the number of dormant neurons (depicted as Dormant), the number of over-active neurons (depicted as Overactive-Number), the sum of NAS (depicted as Overactive-Sum) for over-active neurons over time. (b) Overlap coefficients for Dormant/Over-active neurons between the current iteration and previous iterations. (c) Percentage of dormant neurons that re-enter dormancy after ReDo within different time steps.

we change the number of dormant neurons from 0 to 3. As depicted in Figure 2 (a), with increasing dormant neurons, the mean square error (MSE) loss that fits the target value increases. This indicates that *an increase in dormant neurons leads to reduced expressive power*. Please refer to Appendix D.3 for details.

**TD target non-stationarity exacerbates dormant neurons in MARL.** The TD target in reinforcement learning is non-stationary[38]. In the MARL training process, target networks for mixing networks are typically used to stabilize TD targets. We study the impact of target non-stationarity by varying its update interval, where a smaller interval indicates greater non-stationarity. As analyzed in Figure 1, the dormant neuron phenomenon primarily exists in the mixing network, so we only control the target network of the mixing network, and the comparison focuses on the dormancy ratio in the mixing network. Experimental results for the QMIX method are presented in Figure 2 (b). As depicted, with a smaller update interval, the ratio of dormant neurons increases, indicating that increased non-stationarity in the target network results in a higher presence of dormant neurons in the MARL mixing network.

**The presence of over-active neurons correlate with dormant neurons.** Through careful inspection of the neurons during MARL network training, we observe an interesting phenomenon that has not been discovered before: some neurons exhibit very large normalized activation scores (NAS) throughout the training process. We study the percentage contribution of the average NAS of each neuron to the total average NAS of all neurons. To this end, we examine such percentage in the last layer (with 64 neurons) of QMIX's mixing network in 27m_vs_30m from SMAC. Figure 2 (c) depict the top 25 neurons which have the largest percentage. Neurons whose percentage is over 5% is depicted in red, neurons whose percentage are too low are not plotted, while the other neurons are plotted in green. The results show that most NAS are concentrated on a few neurons, while the NAS of other neurons are relatively low. We refer to these neurons with large NAS as over-active neurons, and define them as follows.

**Definition 4** (over-active neuron). *A neuron $i$ is an over-active neuron if its score $s_i^\ell \geq \beta$ (i.e., 3).*

In Figure 3 (a), the percentage contribution of the numbers of dormant neurons to all neurons, and the percentage contribution of the number of over-active neurons to all neurons, along with the percentage contribution of the sum of NAS for over-active neurons to the sum of NAS of all neurons, are depicted in red, blue and green, respectively. We find that, albeit there are only a few over-active neurons, their NAS takes up a large percentage of the neural network's NAS. This percentage increases steadily with the training process, correlating with the increase of dormant neurons. As the percentage of the over-active neurons' NAS continues to increase, the percentage of NAS for the other neurons decreases. *We conjecture that the presence of over-active neurons impacts the existence of dormant neurons.*

**Dormant/Over-active neurons remain dormant/over-active**. To study the impact of over-active neurons on dormant neurons, we examine the percentage of dormant/over-active neurons that remain dormant/over-active. As depicted in Figure 3 (b), there is a significant overlap among dormant/over-active neurons. The presence of over-active neurons appears to be a significant factor contributing to the dormant neuron phenomenon, which has never been considered in previous studies. We use a parameter perturbing method [4] to periodically recycle the dormant neurons. Then we depict the

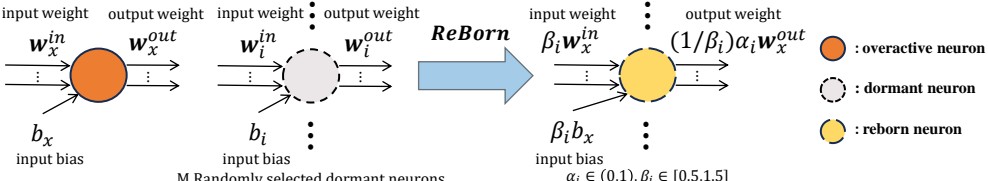

Figure 4: The procedure of ReBorn neurons. The weights of over-active neurons are distributed to M randomly picked dormant neurons.

percentage of dormant neurons that re-enter dormancy within 0.2 Million steps and 0.6 Million steps in Figure 3 (c). As it is depicted in the Figure, there is still a significant overlap among dormant neurons. This indicates that parameter perturbing methods (such as Redo [4]) may not work efficiently for MARL, as it does not consider over-active neurons. We conjecture that this may be due to the fact that methods developed for single-agent RL may change the neural network weights regarding agent cooperation, which lead to *forgetting learned cooperative knowledge that is encoded in neural network*.

## 5 The ReBorn Method

In this section, we describe the Knowledge Invariant Principle, which ensures that learned action preferences do not change after perturbing neurons. We show that methods failing to satisfy this principle could lead to the violation of the individual-Global-Max (IGM) principle, which is important for MARL. Then, we present the ReBorn method, which satisfies the KI principle. It balances the weights among dormant neurons and over-active neurons for the mixing network.

### 5.1 Knowledge Invariant Principle

Multi-agent Reinforcement Learning suffers from the dormant neuron phenomenon and the existence of over-active neurons which make the learning process inefficient. Researchers have proposed several methods [4, 5, 6] that change the weights of neurons. However, these methods overlook the complex interactions among multi-agents, and their learned knowledge may be forgotten after perturbing neurons. We formulate the memorization requirement for learned cooperation knowledge after neuron perturbations as the *Knowledge Invariant Principle*, which is defined as follows.

**Definition 5** (Knowledge Invariant Principle (KI)). *A joint state-action value function is represented as $Q_{tot}^{\theta,\phi}(\boldsymbol{\tau}, \boldsymbol{u}) = f_\theta(Q_1^\phi(\boldsymbol{\tau}_1, u_1), ..., Q_N^\phi(\boldsymbol{\tau}_N, u_N))$, where $f_\theta$ is the mixing function that mixes $Q_i$ into $Q_{tot}$, $\boldsymbol{\tau}$ is joint observation-action history, $\boldsymbol{u} = [u_1, ...u_N]$ is the joint action of multi-agent, $g : \mathbb{R} \mapsto \mathbb{R}$ is a function that maps weights in $\theta$ to $\hat{\theta}$, $g(\theta) = \hat{\theta}$. $h : \mathbb{R} \mapsto \mathbb{R}$, $h(\phi) = \tilde{\phi}$, $h$ map the weights in $\phi$ to $\tilde{\phi}$. If the following condition holds:*

$$Q_{tot}^{\theta,\phi}(\boldsymbol{\tau}, \boldsymbol{u}) \geq Q_{tot}^{\theta,\phi}(\boldsymbol{\tau}, \boldsymbol{u}') \Rightarrow Q_{tot}^{\hat{\theta},\tilde{\phi}}(\boldsymbol{\tau}, \boldsymbol{u}) \geq Q_{tot}^{\hat{\theta},\tilde{\phi}}(\boldsymbol{\tau}, \boldsymbol{u}'), \quad \exists!k : u_k \neq u'_k \quad (4)$$

*then, the two functions $g$ and $h$ satisfy the Knowledge Invariant Principle for $Q_{tot}^{\theta,\phi}$, where $[Q_i(\boldsymbol{\tau}_i, u_i)]_{i=1}^N$ is individual agent utility function, $N$ is the number of agents, $\boldsymbol{\tau}_i$ and $u_i$ are the observation-action history and action of agent $i$, respectively. $\exists!$ represents the concept of **unique existence**.*

Given two functions $g$ and $h$ which satisfy the KI principle, if we use them to change the joint state-action value function $Q_{tot}^{\theta,\phi}$ to $Q_{tot}^{\hat{\theta},\tilde{\phi}}$  $\exists!k : u_k \neq u'_k$, the learned knowledge that $\boldsymbol{u}$ is preferred over $\boldsymbol{u}'$ before applying $g$ and $h$ does not change after applying the two functions. With the KI principle, we show the following theorem.

**Theorem 1.** *Parameter perturbing methods that do not satisfy the Knowledge Invariant (KI) principle cannot guarantee adherence to the Individual-Global-Max (IGM) principle.*

We have theoretically shown that a parameter perturbing method that does not satisfy the KI principle could lead to the *violation of the IGM principle*, which is the most important principle in MARL value factorization methods [2, 9, 20, 10]. Furthermore, we theoretically show that two state-of-the-art RL parameter perturbing methods, Redo [4] and ReSet [5], do not satisfy the KI principle, as detailed in Theorem 2 and Theorem 3, respectively. These theorems and proofs are detailed in Appendix B.

## 5.2 ReBorn: a Weight Sharing Method among Dormant and Over-active Neurons.

To address the issues caused by the dormant neuron phenomenon and the existence of over-active neurons, which reduce network expressivity, we propose ReBorn, a simple but effective method that shares the weights from over-active neurons with dormant neurons.

ReBorn uses an identity function $h(\theta) = \theta$ to map the parameters of agent networks to themselves, and uses function $g(\theta)$ to perturb the parameters $\theta$ of mixing networks. The process of $g(\theta)$ is described as follows. For each over-active neuron $x$, we randomly select $M$ dormant neurons that belong to the same layer as $x$. Here, $M$ is an random integer between 2 to 5. The selected dormant neurons, indexed by $i$, will share weights with neuron $x$. After weight sharing, these neurons will not be selected again. We denote $\boldsymbol{w}_x^{in}$ as the input weights for neuron $x$, $b_x$ as the bias of neuron $x$, $\boldsymbol{w}_x^{out}$ as the output weights. The main procedure of the ReBorn method is depicted in Figure 4.

The input weights of dormant neurons $\boldsymbol{w}_i^{in}$ are reborn as $\beta_i \boldsymbol{w}_x^{in}$, and the input weight of the over-active neuron $x$ becomes $\beta_0 w_x^{in}$. The output weights for neuron $x$ and $i$ are reborn as $\frac{1}{\beta_0} \alpha_0 w_x^{out}$ and $\frac{1}{\beta_i} \alpha_i w_x^{out}$. The biases for the over-active and dormant neurons are set to $\beta_0 b_x$, $\beta_i b_x$. $[\beta_i]_{i=0}^M$ are sampled between 0.5 and 1.5. They are used to ensure more variation among neurons. $[\alpha_i]_{i=0}^M$ is obtained through sampling $M + 1$ from a normal distribution, and then a Softmax operator is performed on them to ensure $\sum_{i=0}^M \alpha_i = 1$. For dormant neurons that are not selected, we use Xavier initialization to reset their weights.

Although ReBorn is simple, we have theoretically demonstrated that it satisfies the KI principle for QMIX through the following theorem.

**Theorem 2.** *ReBorn satisfies the KI principle for the QMIX [2] value factorization method.*

Moreover, we have theoretically shown that ReBorn satisfies the KI principle for a value factorization method: QPLEX in Theorem 4, a distributional value factorization method DMIX in Theorem 5, and a risk-sensitive value factorization method RMIX in Theorem 6. Furthermore, we show that after using ReBorn, the value functions $Q_{tot}$ learned by QMIX, QPLEX, DMIX, and RMIX still satisfy the IGM principle in Corollary 1 to 4. These theorems and proofs are listed in **Appendix B**.

# 6 Empirical Evaluations

In this section, we present experimental results and discuss their implications. We begin with a brief overview of our experimental setup in Section 6.1. Subsequently, we examine ReBorn's robust applicability to various MARL value factorization algorithms in Section 6.2. Furthermore, we demonstrate that ReBorn outperforms other parameter perturbing methods that are extended to MARL in Section 6.3. Lastly, we conduct a series of ablation studies in Section 6.4. All detailed experimental results can be found in Appendix D.4.

## 6.1 Environmental Setup

**Environments.** In our experiments, we employ three distinct environments that challenge the coordination and adaptability of MARL algorithms. **Predator-prey** simulates a grid world where multiple predators collaborate to capture preys dispersed throughout the map. A successful capture requires at least 2 predators to execute the *capture* action simultaneously, posing a great challenge for the algorithm's coordination ability. **The StarCraft Multi-Agent Challenge (SMAC) [13]** is a popular benchmark used extensively in MARL, where multiple ally units controlled by MARL algorithms aim to defeat enemy units controlled by the game's built-in AI. **SMACv2 [14]** features units that are randomly generated and positioned, enhancing stochasticity and significantly increasing the complexity of the scenarios. Please refer to Appendix D.2 for detailed descriptions.

**Baselines and training.** ReBorn, as a parameter perturbation mechanism, is applicable to various value factorization algorithms. We select 4 classical algorithms with different types: QMIX, QPLEX, DMIX and RMIX. ReDo, ReSet, SR [31] and MARR [32] four common parameter perturbation methods in deep RL, are adapted to MARL variants to serve as baselines. Detailed implementations and parameter configurations for each algorithm are available in Appendix D.1.

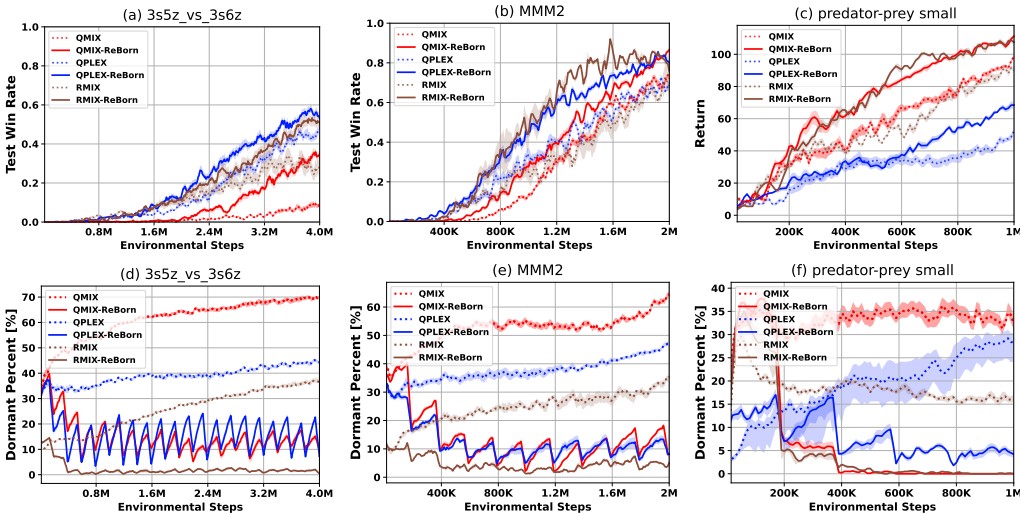

Figure 5: ReBorn can improve the performance of various value factorization algorithms: (a-b) the test win rate for the 3s5z_vs_3s6z and the MMM2 environments, (c) the return for predator-prey small environment, (d-f) the dormant percent for the the 3s5z_vs_3s6z, the MMM2, and the predator-prey small environment.

## 6.2 ReBorn can improve the performance of various value factorization algorithms

In this section, we investigate the applicability of ReBorn through validating ReBorn's ability to enhance performance across various value factorization algorithms (QMIX, QPLEX, RMIX) in different experimental scenarios (3s5z_vs_3s6z, MMM2, predator-prey small). According to the experimental results presented in Figures 5, ReBorn can improve the performance of multiple algorithms and effectively reduce the dormant ratio of the mixing networks in diverse settings. More detailed experimental results can be found in Appendix D.4.1 and Appendix D.4.5.

## 6.3 ReBorn is superior to other RL parameter perturbing methods

We explore the superiority of ReBorn by applying different parameter perturbing methods to QMIX across various experimental scenarios (MMM2, 27m_vs_30m, predator-prey large). We added ReDo, Reset, SR and MARR for comparison and further analyzed the dormant ratios and the over-active sum ratios. ReDo and ReSet are common parameter perturbation methods in deep RL, while SR and MARR are reset replay methods. All of them are adapted to MARL variants to serve as baselines.

The results depicted in Figure 6 illustrate the win rates, the dormant ratios and the over-active sum ratios (the ratio of the sum of normalized activation scores of over-active neurons to the total sum of scores of all neurons) across different scenarios. The results indicate that compared to ReDo and ReSet, ReBorn can further enhance algorithm's performance and more effectively reduce both the dormant and over-active sum ratios of the mixing network. Please refer to Appendix D.4.4 for more experimental results.

## 6.4 Ablation Study and Discussion

### 6.4.1 Satisfying the KI Principle is of great importance

In this section, we demonstrate the importance of adhering to the Knowledge Invariance (KI) principle. Our analysis in Appendix B shows that applying ReBorn only to the mixing network adheres to the KI principle, while using it on the entire network results in a violation. We compared the performance of value factorization algorithms under the MMM2 scenario in SMAC, focusing on those that either adhere to or violate the KI principle. As illustrated in Figure 7, maintaining KI with ReBorn enhances the performance across all baseline algorithms, whereas violating it leads to performance drop in QMIX and QPLEX, highlighting the great importance of satisfying the KI principle.

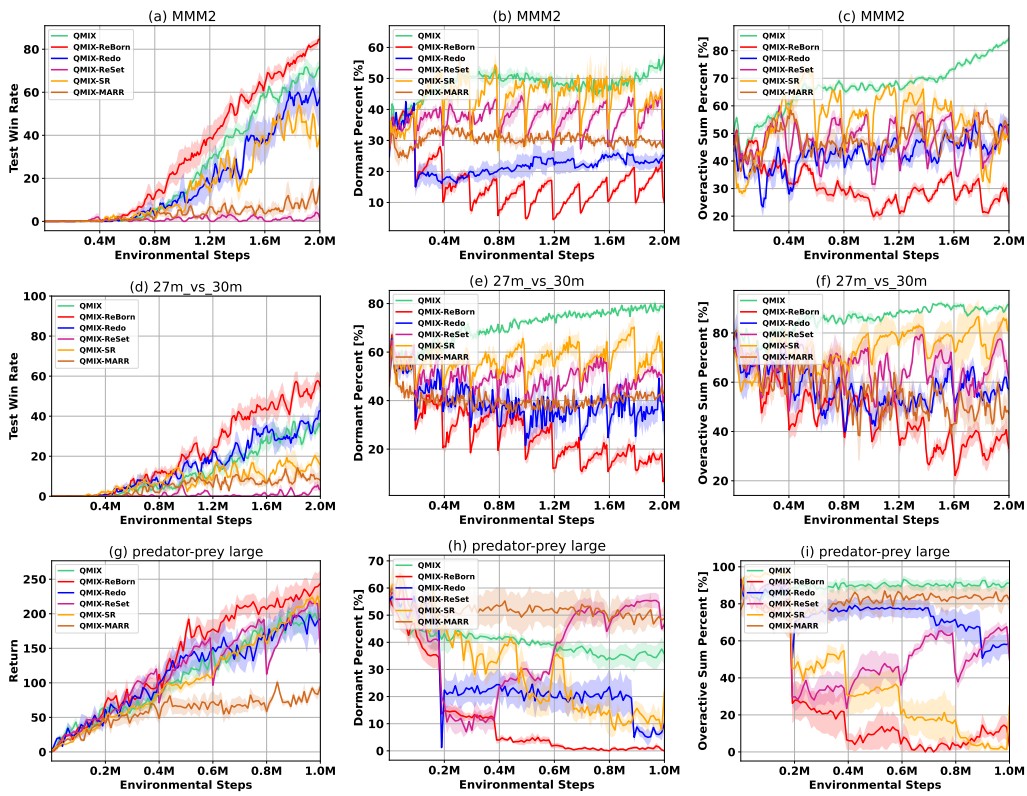

Figure 6: Comparison with other Parameter Perturbing Methods: (a-c) The test win rate, the dormant percentage and the percentage of the sum of normalized activation score (NAS) for the MMM2 environment. (d-f) The test win rate, the dormant percentage, and the percentage of the sum of NAS for the 27m_vs_30m environment. (g-i) The return, the dormant percentage, and the percentage of the sum of NAS for the predator-prey large environment.

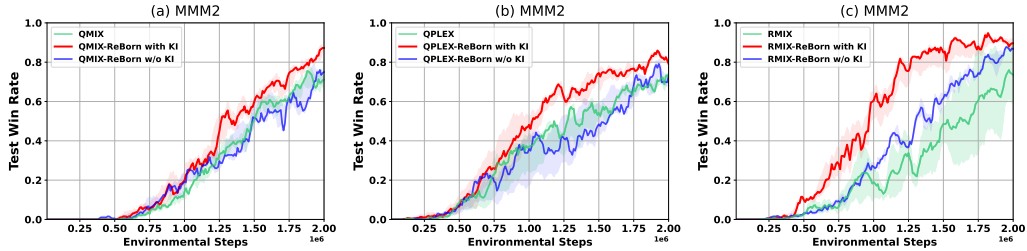

Figure 7: Importance of satisfying the KI Principle for (a) QMIX, (b) QPLEX, and (C) RMIX. A variant of ReBorn without satisfying the KI Principle is depicted as Reborn w/o KI.

### 6.4.2 ReBorn is better than other methods that satisfy the KI principle

In this section, we explore various forms of the weight perturbation function $g(\theta)$ in ReBorn based on QMIX, while keeping the function $h(\theta) = \theta$ constant. In ReBorn, $g(\theta)$ transfers weights from over-active neurons to dormant neurons. In ReBorn (ReDo), $g(\theta)$ periodically re-initializes the weights of dormant neurons. In ReBorn (ReSet), $g(\theta)$ periodically resets the parameters of the last layer of the neural network. In ReBorn (Reverse ReDo), $g(\theta)$ periodically resets the input and output weights of over-active neurons. In ReBorn (Pruning), $g(\theta)$ periodically prunes dormant neurons. To ensure the accuracy of our conclusions, we conduct experiments across various value factorization algorithms. The experimental results shown in Figure 8 demonstrate that compared with other methods, ReBorn significantly enhance the performance of value factorization algorithms.

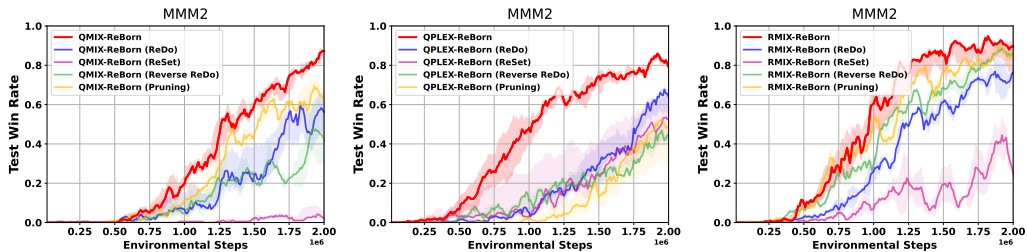

Figure 8: Comparison with other methods that satisfy the KI principle for (a) QMIX, (b) QPLEX, and (C) RMIX.

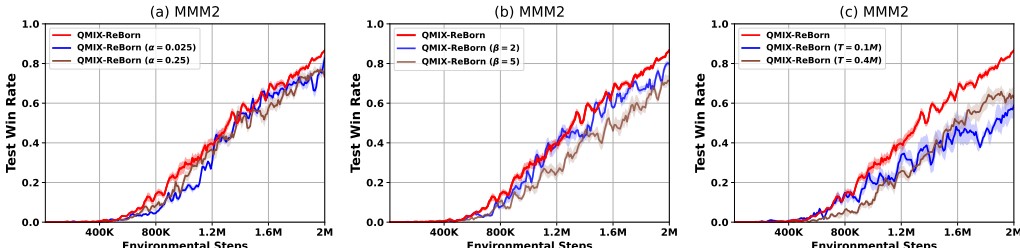

Figure 9: Ablation of different hyperparameters in ReBorn. (a) the dormant threshold $\alpha$. (b) the over-active threshold $\beta$. (c) the ReBorn interval $T$.

### 6.4.3 Sensitivity analyses of hyper-parameters

The ablation study in Figure 9 illustrates the impact of different hyperparameter settings in QMIX-ReBorn, focusing on the ablation of the dormant threshold $\alpha$, the over-active threshold $\beta$, and the ReBorn interval $T$. The default configuration of QMIX-ReBorn is $\alpha = 0.1$, $\beta = 3$, and $T = 0.2M$. We modify each hyperparameter individually, and the experimental results indicate that appropriate hyperparameters help to better balance network activation, thereby enhancing overall performance.

## 7 Conclusion

In this work, we identify the dormant neuron phenomenon in Multi-Agent Reinforcement Learning (MARL) Value Factorization. Such a phenomenon mainly exists in the mixing network, which hurts its expressive ability. We discover the existence of over-active neurons, which correlate with dormant neurons. Existing parameter perturbing methods do not work efficiently for the dormant neurons in MARL, due to the ignorance of over-active neurons and the forgotten of learned knowledge. We formulate the memorization requirement for learning agents' cooperation knowledge as the Knowledge Invariant (KI) principle. In this work, we propose ReBorn, which is a simple but effective parameter perturbing method. We show that it satisfies the KI principle and can improve the performance of multiple value factorization methods better than other parameter perturbing methods.

**Acknowledgement**   This work was partially supported by the Fundamental Research Funds for the Central Universities (No. 20720230033), by PDL (2022-PDL-12). We would like thank the anonymous reviewers for their valuable suggestions.

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

# Appendix

## A  Background

### A.1  Dec-POMDPs

We consider Decentralized Partially Observable Markov Decision Processes (Dec-POMDPs) [16] in modeling cooperative multi-agent reinforcement learning (MARL) scenarios. A Dec-POMDP can be formally described by the tuple $G = \langle \mathcal{S}, \{\mathcal{U}_i\}_{i=1}^N, P, r, \{\mathcal{O}_i\}_{i=1}^N, \{\sigma_i\}_{i=1}^N, N, \gamma \rangle$, where $\mathcal{N}$ represents the set of agents, $\mathcal{S}$ is a finite set of states, and $\mathcal{U}_i$ is the set of actions available to agent $i$. At time step $t$, each agent $i$ chooses an action $u_i^t \in \mathcal{U}_i$, forming a joint action $\boldsymbol{u}^t \in \mathcal{U}^N = \mathcal{U}_1 \times \ldots \times \mathcal{U}_N$. This leads to a transition to a new state $s^{t+1} \sim P(\cdot|s^t, \boldsymbol{u}^t)$ and a joint reward $r^t$.. In consideration of partial observability, each agent can only access an individual observation $o_i^t \in O_i$, which is drawn from $o_i^t \sim \sigma^i(\cdot|s^t)$. $\gamma$ denotes the discounting factor. Each agent acts base on individual policy $\pi_i(u_i|\tau_i)$, $\tau_i = (O_i \times U_i)^*$ represents agent's local action-observation history. the global action-observation history is denoted as $\tau \in \mathcal{T}^N := \tau_1 \times \ldots \times \tau_N$, on which it conditions the joint policy $\pi = <\pi_1, \ldots, \pi_N>$. The joint policy $\pi$ has a joint action-value function: $Q^\pi(s_t, \mathbf{u}_t) = \mathbb{E}_{s_{t+1:\infty}, \mathbf{u}_{t+1:\infty}} [R_t \mid s_t, \mathbf{u}_t]$, where $R_t = \sum_{i=0}^\infty \gamma^i r_{t+i}$ is the discounted return.

### A.2  Value Function Factorization

For cooperative multi-agent reinforcement learning tasks with partial observability challenges, agents are supposed to select actions solely based on their local observations. This presents significant challenges to global coordination in scenarios where communication is unavailable. To efficiently solve this problem, centralized training with decentralized execution (CTDE) was proposed as a popular paradigm. During centralized training, access to global information is available, while only local action-observation histories are accessible during decentralized execution phase. Value factorization is a class of effective value-based methods under the CTDE paradigm, where agents make decisions based on individual utility functions. The mixing network is employed during training to fit the relationship between the joint value function and individual utility functions. Among all value factorization methods, the Individual-Global-Max (IGM) principle proposed by [9] is a critical criterion that must be adhered to, ensuring the consistency between local and joint optimal action selections. The definition of the IGM principle is as follows:

**Definition 6** (IGM). *For a joint state-action value function $Q_{jt} : \mathcal{T}^N \times \mathcal{U}^N \mapsto \mathbb{R}$, where $\tau \in \mathcal{T}^N$ is a joint action-observation history and $\mathbf{u}$ is the joint action, if there exists individual state-action functions $[Q_i : \mathcal{T}_i \times \mathcal{U}_i \mapsto \mathbb{R}]_{i=1}^N$, such that the following conditions are satisfied*

$$\arg\max_{\mathbf{u}} Q_{jt}(\boldsymbol{\tau}, \boldsymbol{u}) = (\arg\max_{u_1} Q_1(\tau_1, u_1), \ldots, \arg\max_{u_n} Q_N(\tau_N, u_N)), \qquad \text{(A.1)}$$

*then, we can state that $[Q_i]_{i=1}^N$ satisfy IGM for $Q_{jt}$ under $\tau$, or $Q_{jt}(\boldsymbol{\tau}, \boldsymbol{u})$ is factorized by $[Q_i(\tau_i, u_i)]_{i=1}^N$.*

In recent years, ensuring the adherence to the IGM principle, a series of value factorization methods have been proposed. VDN imposes additive constraints on the mixing network, and QMIX enhances VDN's representation ability by imposing monotonicity constraints. These constraints are sufficient conditions for IGM, limiting the representational ability of the joint value function. QTRAN transforms the IGM principle into a linear constraint and proposes an easily factorizable form. Qatten uses the attention mechanism to model each agent's impact on the global situation. QPLEX decomposes the state-action value function into a state value part and an advantage value part. ResQ converts the joint value function into the sum of a main function and a residual function, deriving optimal policy through masking.

# B  Principle and Theorem

**Theorem 1.** *After the ReBorn process, the learned value function of the QMIX [2] value factorization method still satisfy the KI principle.*

$$Q_{tot}^{\theta,\phi}(\boldsymbol{\tau},\boldsymbol{u}) = f_\theta(Q_1^\phi(\boldsymbol{\tau}_1,u_1),...,Q_N^\phi(\boldsymbol{\tau}_N,u_N)) + V_\theta(\tau) \quad \frac{\partial f}{\partial Q_i^\phi} \geq 0 \tag{B.2}$$

$$h(w) = w, \quad \forall w \in \phi \quad h \text{ is an identity function} \tag{B.3}$$

$$g(w) = \begin{cases} \beta_i w_x^{in} & \text{input weights of dormant neurons } \boldsymbol{w}_i^{in} \\ \beta_0 w_x^{in} & \text{input weights of over-active neurons } \boldsymbol{w}_x^{in} \\ \frac{1}{\beta_i}\alpha_i w_x^{out} & \text{output weights of dormant neurons } \boldsymbol{w}_i^{out} \\ \frac{1}{\beta_0}\alpha_0 w_x^{out} & \text{output weights of over-active neurons } \boldsymbol{w}_x^{out} \\ \beta_0 b_x & \text{bias of over-active neurons } \boldsymbol{b}_x \\ \beta_i b_x & \text{bias of dormant neurons } \boldsymbol{b}_i \\ Xavier(w) & \text{weights of non-select dormant neurons} \\ w & \text{otherwise} \end{cases} \tag{B.4}$$

where $Q_{tot}^{\theta,\phi}(\boldsymbol{\tau},\boldsymbol{u}) = f_\theta(Q_1^\phi(\boldsymbol{\tau}_1,u_1),...,Q_N^\phi(\boldsymbol{\tau}_N,u_N))$ is the joint state-action value funtion, $f_\theta$ is the value factorization function of QMIX. In Reborn, g maps the parameters $\theta$ of the mixing network and the parameters $\phi$ of the agent network to $\hat{\theta}$.

*Proof.* Through the use of non-negative activation function (e.g. absolute) and hypernet [39], QMIX can ensure the following property.

$$\frac{\partial f_\theta}{\partial Q_i^\phi} \geq 0 \quad \forall \theta \text{ monotonicity property} \tag{B.5}$$

It indicates that if $Q_i^\phi$ increase, then the value of $f$ increases.

$$Q_{tot}^{\theta,\phi}(\boldsymbol{\tau},\boldsymbol{u}) \geq Q_{tot}^{\theta,\phi}(\boldsymbol{\tau},\boldsymbol{u}'), \exists!k : u_k \neq u_k' \tag{B.6}$$

$$Q_{tot}^{\theta,\phi}(\boldsymbol{\tau},[u_1,...,u_N]) \geq Q_{tot}^{\theta,\phi}(\boldsymbol{\tau},[u_1',...,u_N']) \quad \text{expand } \boldsymbol{u} \tag{B.7}$$

$$f_\theta(Q_1^\phi(\boldsymbol{\tau}_1,u_1),...,Q_N^\phi(\boldsymbol{\tau}_N,u_N)) + V_\theta(\tau) \geq f_\theta(Q_1^\phi(\boldsymbol{\tau}_1,u_1'),...,Q_N^\phi(\boldsymbol{\tau}_N,u_N')) + V_\theta(\tau) \tag{B.8}$$

$$f_\theta(Q_1^\phi(\boldsymbol{\tau}_1,u_1),...,Q_N^\phi(\boldsymbol{\tau}_N,u_N)) \geq f_\theta(Q_1^\phi(\boldsymbol{\tau}_1,u_1'),...,Q_N^\phi(\boldsymbol{\tau}_N,u_N')) \tag{B.9}$$

$$Q_k^\phi(\boldsymbol{\tau}_k,u_k) \geq Q_k^\phi(\boldsymbol{\tau}_k,u_k') \quad , \exists!k : u_k \neq u_k' \text{because of (B.5)} \tag{B.10}$$

$$f_{\hat{\theta}}(Q_1^\phi(\boldsymbol{\tau}_1,u_1),...,Q_N^\phi(\boldsymbol{\tau}_N,u_N)) \geq f_{\hat{\theta}}(Q_1^\phi(\boldsymbol{\tau}_1,u_1'),...,Q_N^\phi(\boldsymbol{\tau}_N,u_N')) \tag{B.11}$$

$$f_{\hat{\theta}}(Q_1^\phi(\boldsymbol{\tau}_1,u_1),...,Q_N^\phi(\boldsymbol{\tau}_N,u_N)) + V_{\hat{\theta}}(\tau) \geq f_{\hat{\theta}}(Q_1^\phi(\boldsymbol{\tau}_1,u_1'),...,Q_N^\phi(\boldsymbol{\tau}_N,u_N')) + V_{\hat{\theta}}(\tau) \tag{B.12}$$

$$Q_{tot}^{\hat{\theta},\phi}(\boldsymbol{\tau},\boldsymbol{u}) \geq Q_{tot}^{\hat{\theta},\phi}(\boldsymbol{\tau},\boldsymbol{u}') \tag{B.13}$$

(B.9) to (B.10) is because $u_i = u_i'$, $\forall i \neq i$ , and $u_k \neq u_k'$ and the monotonicy conditions. (B.10) to (B.11) is due to the monotonicy condition, because $u_i = u_i'$, $\forall i \neq i$ , and $u_k \neq u_k'$. Thus, we show that after the ReBorn process, the learned action preference of QMIX does not change. □

**Corollary 1.** *After the ReBorn Process, the value function of QMIX remain satisfies the IGM principle.*

*Proof.* To prove this Corollary is equal to prove that the maximal action remain the same after the ReBorn method. It is shows that the ReBorn method satisfy the KI principle for QMIX, thus the

following condition is satisfy.

$$Q_{tot}^{\theta,\phi}(\boldsymbol{\tau},\boldsymbol{u}) \geq Q_{tot}^{\theta,\phi}(\boldsymbol{\tau},\boldsymbol{u}') \rightarrow Q_{tot}^{\hat{\theta},\phi}(\boldsymbol{\tau},\boldsymbol{u}) \geq Q_{tot}^{\hat{\theta},\phi}(\boldsymbol{\tau},\boldsymbol{u}') \quad , \exists! k : u_k \neq u_k' \tag{B.14}$$

$$Q_{tot}^{\theta,\phi}(\boldsymbol{\tau},\bar{\boldsymbol{u}}) \geq Q_{tot}^{\theta,\phi}(\boldsymbol{\tau},\boldsymbol{u}') \quad \bar{\boldsymbol{u}} = \arg\max_{\boldsymbol{u}} Q_{tot}^{\theta,\phi}(\boldsymbol{\tau},\boldsymbol{u}), \forall \boldsymbol{u}' \tag{B.15}$$

$$\bar{\boldsymbol{u}} = [\bar{u}_1,...,\bar{u}_N] \quad \bar{u}_i = \arg\max_{u_i} Q_i^{\phi}(\boldsymbol{\tau}_i, u_i) \text{ IGM Principle} \tag{B.16}$$

$$Q_{tot}^{\theta,\phi}(\boldsymbol{\tau},\bar{\boldsymbol{u}}) \geq Q_{tot}^{\theta,\phi}(\boldsymbol{\tau},\boldsymbol{u}') \rightarrow Q_{tot}^{\hat{\theta},\phi}(\boldsymbol{\tau},\bar{\boldsymbol{u}}) \geq Q_{tot}^{\hat{\theta},\phi}(\boldsymbol{\tau},\boldsymbol{u}') \forall \boldsymbol{u}' \quad \text{KI Principle} \tag{B.17}$$

$$Q_{tot}^{\hat{\theta},\phi}(\boldsymbol{\tau},\bar{\boldsymbol{u}}) \geq Q_{tot}^{\hat{\theta},\phi}(\boldsymbol{\tau},\boldsymbol{u}'), \forall \boldsymbol{u}' \tag{B.18}$$

$$\bar{\boldsymbol{u}} = \arg\max_{\boldsymbol{u}} Q_{tot}^{\hat{\theta},\phi}(\boldsymbol{\tau},\boldsymbol{u}) = \arg\max_{\boldsymbol{u}} Q_{tot}^{\theta,\phi}(\boldsymbol{\tau},\boldsymbol{u}) \tag{B.19}$$

(B.19) shows that the IGM principle is still preserve after applying ReBorn on the joint state-action value $Q_{tot}$ of QMIX. $\qquad\square$

**Theorem 2.** *ReDo [4] with function g and h does not guarantee satisfying the KI principle for the QMIX [2] value factorization method which is defined as follows.*

$$Q_{tot}^{\theta,\phi}(\boldsymbol{\tau},\boldsymbol{u}) = f_\theta(Q_1^{\phi}(\boldsymbol{\tau}_1,u_1),...,Q_N^{\phi}(\boldsymbol{\tau}_N,u_N)) + V_\theta(\tau) \quad \frac{\partial f}{\partial Q_i^{\phi}} \geq 0 \tag{B.20}$$

$$g(w) = \begin{cases} 0 & w \in \theta_d^o \quad \textit{output weights of dormant neurons} \\ \textit{Xavier(w)} & w \in \theta_d^i \quad \textit{input weights of dormant neurons} \\ w & \textit{otherwise} \end{cases} \tag{B.21}$$

$$h(w) = \begin{cases} 0 & w \in \phi_d^o \quad \textit{output weights of dormant neurons} \\ \textit{Xavier(w)} & w \in \phi_d^i \quad \textit{input weights of dormant neurons} \\ w & \textit{otherwise} \end{cases} \tag{B.22}$$

*where $f_\theta$ is the value factorization function of QMIX, g map the parameters $\theta$ of the mixing network to $\hat{\theta}$, h map the parameters $\phi$ of the agent network to $\tilde{\phi}$, $Xavier(w)$ indicates the Xavier initialization function, $\theta_d^o$, $\theta_d^i$ are the output/input weights of dormant neurons in $\theta$, respectively, $\phi_d^o/\phi_d^i$ are the output/input weights of dormant neurons in $\phi$, respectively.*

*Proof.* We prove this theorem by providing an example that ReDo does not satisfy the KI principle. We assume that the mixing network, parameterized by $\theta$ is a three layer neural work. As it is depicted in Figure 1, the input layer neurons are used for joint state-action history $\tau$. It also takes actions $\boldsymbol{u}$ as input. There are in total four joint actions represent as $\boldsymbol{u}^1 = [0,0], \boldsymbol{u}^2 = [0,1], \boldsymbol{u}^3 = [1,0], \boldsymbol{u}^4 = [1,1]$. There are two agents, each has two actions represent as 0 and 1, respectively. The action of the first/second agent is fed into the second/third neuron of the input layer. The weights of each neurons are marked on the edges and we assume bias $b = 0$. According to the definition 2 and the weights, the blue neuron is a dormant neuron. Assuming $\tau = 1$, we can obtain the relationship of $Q(\boldsymbol{\tau},\boldsymbol{u})$ corresponding to each action:

$$Q(\tau,\boldsymbol{u}^3) > Q(\tau,\boldsymbol{u}^4) > Q(\tau,\boldsymbol{u}^1) > Q(\tau,\boldsymbol{u}^2) \tag{B.23}$$

For the dormant neuron (colored), ReDo B.22 reinitialized the input weights of the neuron using Xavier initialization, the output weights of the neurons are set to zero. The weights after ReDo are as depicted in the right part of Figure 1, and we assume that the bias for each neuron is zero. We can obtain the relationship of $Q'(\boldsymbol{\tau},\boldsymbol{u})$ corresponding to each joint action:

$$Q(\tau,\boldsymbol{u}^4) > Q(\tau,\boldsymbol{u}^3) > Q(\tau,\boldsymbol{u}^2) > Q(\tau,\boldsymbol{u}^1) \tag{B.24}$$

Since the original optimal action $u^3$ in B.23 is different from the one $u^4$ in B.24 after Redo, we can draw a conclusion that *ReDo does not satisfy the KI principle.*

$\qquad\square$

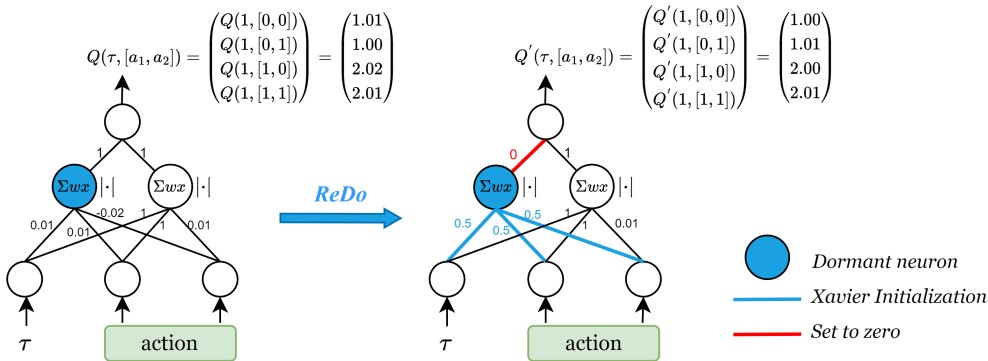

Figure 1: An example to show that ReDo does not satisfy KI principle.

**Theorem 3.** *ReSet [6] with function $g$ and $h$ does not guarantee satisfying the KI principle for the QMIX [2] value factorization method which is defined as follows.*

$$Q_{tot}^{\theta,\phi}(\boldsymbol{\tau},\boldsymbol{u}) = f_\theta(Q_1^\phi(\boldsymbol{\tau}_1,u_1),...,Q_N^\phi(\boldsymbol{\tau}_N,u_N)) + V_\theta(\tau) \quad \frac{\partial f}{\partial Q_i^\phi} \geq 0 \tag{B.25}$$

$$g(w) = \begin{cases} Xavier(w) & w \in \theta \quad w \in \text{weights of the last layer of neural network} \\ w & w \in \text{other layers} \end{cases} \tag{B.26}$$

$$h(w) = \begin{cases} Xavier(w) & w \in \phi \quad w \in \text{weights of the last layer of neural network} \\ w & w \in \text{other layers} \end{cases} \tag{B.27}$$

*where $f_\theta$ is the value factorization function of QMIX, $g$ map the parameters $\theta$ of the mixing network to $\hat{\theta}$, $h$ map the parameters $\phi$ of the agent network to $\tilde{\phi}$, $Xavier(w)$ indicates the Xavier initialization function.*

*Proof.* We prove this theorem by providing an example that ReSet does not satisfy the KI principle. Consider a three-layer mixing network parameterized by $\theta$, which takes joint action-observation history, represented as $\tau$, and actions $\boldsymbol{u}$ as input, shown in Figure 2. There are in total four actions represent as $\boldsymbol{u}^1 = [0,0], \boldsymbol{u}^2 = [0,1], \boldsymbol{u}^3 = [1,0], \boldsymbol{u}^4 = [1,1]$. There are two agents, each has two actions represent as 0 and 1, respectively. The action of the first/second agent is fed into the second/third neuron of the input layer. The weights of each neurons are marked on the edges and we assume bias $b = 0$. Assuming $\tau = 1$, we can obtain the relationship of $Q(\boldsymbol{\tau},\boldsymbol{u})$ corresponding to each action:

$$Q(\tau,\boldsymbol{u}^3) > Q(\tau,\boldsymbol{u}^4) > Q(\tau,\boldsymbol{u}^1) > Q(\tau,\boldsymbol{u}^2) \tag{B.28}$$

The weights of the last layer are reinitialized using Xavier initialization according to B.27. The weights after ReDo are as depicted in the right part of Figure 2. We can obtain the relationship of $Q'(\boldsymbol{\tau},\boldsymbol{u})$ corresponding to each joint action:

$$Q(\tau,\boldsymbol{u}^4) = Q(\tau,\boldsymbol{u}^3) > Q(\tau,\boldsymbol{u}^2) = Q(\tau,\boldsymbol{u}^1) \tag{B.29}$$

Since the original optimal action $u^3$ in B.28 is different from the optimal action $u^4$ in B.29 after Reset, we can draw a conclusion that *Reset does not satisfy the KI principle*.

$\square$

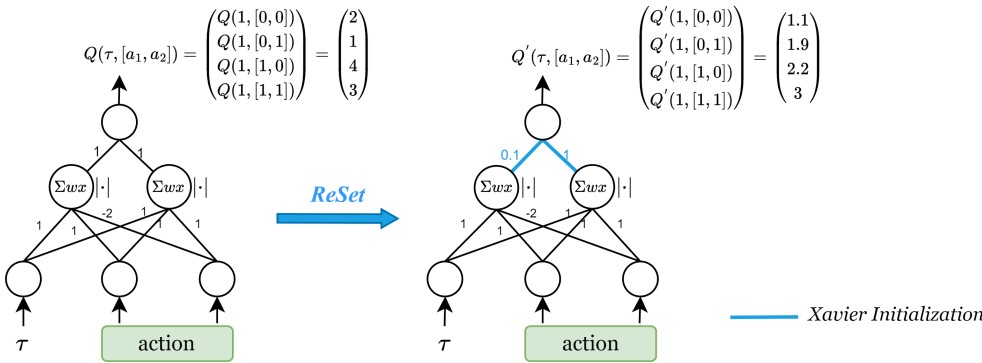

Figure 2: An example to show that Reset does not satisfy KI principle.

**Theorem 4.** *ReBorn with functions $g$ and $h$ satisfies the KI principle for the QPLEX [10] value factorization method.*

$$Q_{tot}^{\theta,\phi}(\boldsymbol{\tau}, \boldsymbol{u}) = V_{tot}^{\theta,\phi}(\boldsymbol{\tau}) + A_{tot}^{\theta,\phi}(\boldsymbol{\tau}, \boldsymbol{u}) \tag{B.30}$$

$$A_{tot}^{\theta,\phi}(\boldsymbol{\tau}, \boldsymbol{u}) = f_\theta(A_1^\phi(\boldsymbol{\tau}_1, u_1), ..., A_N^\phi(\boldsymbol{\tau}_N, u_N)) \quad \frac{\partial f}{\partial A_i^\phi} \geq 0 \tag{B.31}$$

$$Q_i^\phi(\boldsymbol{\tau}_i, \boldsymbol{u}_i) = A_i^\phi(\boldsymbol{\tau}_i, u_i) + V_i^\phi(\boldsymbol{\tau}_i) \quad V_i^\phi(\boldsymbol{\tau}_i) = \max_{u_i} Q_i^\phi(\boldsymbol{\tau}_i, u_i) \tag{B.32}$$

$$V_{tot}^{\theta\,\phi}(\boldsymbol{\tau}) = \max_{\boldsymbol{u}} Q_{tot}^{\theta,\phi}(\boldsymbol{\tau}, \boldsymbol{u}) \tag{B.33}$$

$$g(w) = \begin{cases} \beta_i w_x^{in} & \text{input weights of dormant neurons } \boldsymbol{w}_i^{in} \\ \beta_0 w_x^{in} & \text{input weights of over-active neurons } \boldsymbol{w}_x^{in} \\ \frac{1}{\beta_i}\alpha_i w_x^{out} & \text{output weights of dormant neurons } \boldsymbol{w}_i^{out} \\ \frac{1}{\beta_0}\alpha_0 w_x^{out} & \text{output weights of over-active neurons } \boldsymbol{w}_x^{out} \\ \beta_0 b_x & \text{bias of over-active neurons } \boldsymbol{b}_x \\ \beta_i b_x & \text{bias of dormant neurons } \boldsymbol{b}_i \\ Xavier(w) & \text{weights of non-select dormant neurons} \\ w & \text{otherwise} \end{cases} \tag{B.34}$$

$$h(w) = w, \quad \forall w \in \phi \quad h \text{ is an identity function} \tag{B.35}$$

*where $f_\theta$ is the value factorization function of QMIX. In Reborn, $g$ map the parameters $\theta$ of the mixing network to $\hat{\theta}$.*

*Proof.* QPLEX uses non-negative weighted attention network to implement $f_\theta$ which satisfies the following property.

$$\frac{\partial f_\theta}{\partial A_i^\phi} \geq 0 \quad \forall i, \ \forall \theta, \ \forall \phi \text{ monotonicity property} \tag{B.36}$$

$$Q_{tot}^{\theta,\phi}(\boldsymbol{\tau},\boldsymbol{u}) \geq Q_{tot}^{\theta,\phi}(\boldsymbol{\tau},\boldsymbol{u}') \quad \exists!k : u_k \neq u_k' \tag{B.37}$$

$$V_{tot}^{\theta,\phi}(\boldsymbol{\tau}) + A_{tot}^{\theta,\phi}(\boldsymbol{\tau},\boldsymbol{u}) \geq V_{tot}^{\theta,\phi}(\boldsymbol{\tau}) + A_{tot}^{\theta,\phi}(\boldsymbol{\tau},\boldsymbol{u}') \tag{B.38}$$

$$V_{tot}^{\theta,\phi}(\boldsymbol{\tau}) + A_{tot}^{\theta,\phi}(\boldsymbol{\tau},[u_1,...,u_N]) \geq V_{tot}^{\theta,\phi}(\boldsymbol{\tau}) + + A_{tot}^{\theta,\phi}(\boldsymbol{\tau},[u_1',...,u_N']) \tag{B.39}$$

$$V_{tot}^{\theta,\phi}(\boldsymbol{\tau}) + f_\theta(A_1^\phi(\boldsymbol{\tau}_1,u_1),...,A_N^\phi(\boldsymbol{\tau}_N,u_N)) \geq V_{tot}^{\theta,\phi}(\boldsymbol{\tau}) + f_\theta(A_1^\phi(\boldsymbol{\tau}_1,u_1'),...,A_N^\phi(\boldsymbol{\tau}_N,u_N')) \tag{B.40}$$

$$f_\theta(A_1^\phi(\boldsymbol{\tau}_1,u_1),...,A_N^\phi(\boldsymbol{\tau}_N,u_N)) \geq f_\theta(A_1^\phi(\boldsymbol{\tau}_1,u_1'),...,A_N^\phi(\boldsymbol{\tau}_N,u_N')) \tag{B.41}$$

$$A_k^\phi(\boldsymbol{\tau}_k,u_k)) \geq A_k^\phi(\boldsymbol{\tau}_k,u_k')) \tag{B.42}$$

$$f_{\hat{\theta}}(A_1^\phi(\boldsymbol{\tau}_1,u_1),...,A_N^\phi(\boldsymbol{\tau}_N,u_N)) \geq f_{\hat{\theta}}(A_1^\phi(\boldsymbol{\tau}_1,u_1'),...,A_N^\phi(\boldsymbol{\tau}_N,u_N')) \tag{B.43}$$

$$V_{tot}^{\hat{\theta},\phi}(\boldsymbol{\tau}) + f_{\hat{\theta}}(Q_1^\phi(\boldsymbol{\tau}_1,u_1),...,Q_N^\phi(\boldsymbol{\tau}_N,u_N)) \geq V_{tot}^{\hat{\theta},\phi}(\boldsymbol{\tau}) + f_{\hat{\theta}}(Q_1^\phi(\boldsymbol{\tau}_1,u_1'),...,Q_N^\phi(\boldsymbol{\tau}_N,u_N')) \tag{B.44}$$

$$Q_{tot}^{\hat{\theta},\phi}(\boldsymbol{\tau},\boldsymbol{u}) \geq Q_{tot}^{\hat{\theta},\phi}(\boldsymbol{\tau},\boldsymbol{u}') \tag{B.45}$$

(B.41) to (B.42) is because $u_i = u_i'$, $\forall i \neq k$ , and $u_k \neq u_k'$ and the monotonicity conditions. (B.42) to (B.43) is due to the monotonicity condition, because $u_i = u_i' \ \forall i \neq i$ , and $u_k \neq u_k'$. Thus, we show that after the ReBorn process, the learned action preference of QPLEX does not change. $\square$

**Corollary 2.** *After the ReBorn Process, the value function of QPLEX remain satisfies the IGM principle.*

*Proof.* The proof is the same as the proof for showing after the ReBorn process, QMIX satisfies the IGM principle. It is omitted for brevity. $\square$

**Theorem 5.** *ReBorn with functions $g$ and $h$ satisfies the KI principle for the DMIX [10] value factorization method. DMIX is a distribution MARL algorithm which models the distributional return $Z_{tot}$ of multi-agent system.*

$$Z_{tot}^{\theta,\phi}(\boldsymbol{\tau},\boldsymbol{u}) = Z_{mean}^{\theta,\phi}(\boldsymbol{\tau},\boldsymbol{u}) + Z_{shape}^{\theta,\phi}(\boldsymbol{\tau},\boldsymbol{u}) \tag{B.46}$$

$$Q_{tot}^{\theta,\phi}(\boldsymbol{\tau},\boldsymbol{u}) = Z_{mean}^{\theta,\phi}(\boldsymbol{\tau},\boldsymbol{u}) = f_\theta(Q_1^\phi(\boldsymbol{\tau}_1,u_1),...,Q_N^\phi(\boldsymbol{\tau}_N,u_N)) + V_\theta(\tau) \quad \frac{\partial f}{\partial Q_i^\phi} \geq 0 \tag{B.47}$$

$$Q_i^\phi(\boldsymbol{\tau}_i,\boldsymbol{u}_i) = \mathbb{E}[Z_i^\phi(\boldsymbol{\tau}_i,u_i)] \text{ expectation over possible outcome of } Z_i \tag{B.48}$$

$$Z_{shape}^{\theta\phi}(\boldsymbol{\tau}_i,\boldsymbol{u}_i) = \sum_{i=1}^N (Z_i^\phi(\boldsymbol{\tau}_i,\boldsymbol{u}_i) - Q_i^\phi(\boldsymbol{\tau}_i,\boldsymbol{u}_i)) \tag{B.49}$$

$$h(w) = w, \quad \forall w \in \phi \quad h \text{ is an identity function} \tag{B.50}$$

$$g(w) = \begin{cases} \beta_i\alpha_i w_x^{in} & \text{input weights of dormant neurons } \boldsymbol{w}_i^{in} \\ \beta_0\alpha_0 w_x^{in} & \text{input weights of over-active neurons } \boldsymbol{w}_x^{in} \\ \frac{1}{\beta_i} w_x^{out} & \text{output weights of dormant neurons } \boldsymbol{w}_i^{out} \\ \frac{1}{\beta_0} w_x^{out} & \text{output weights of over-active neurons } \boldsymbol{w}_x^{out} \\ \beta_0 b_x & \text{bias of over-active neurons } \boldsymbol{b}_x \\ \beta_i b_x & \text{bias of dormant neurons } \boldsymbol{b}_i \\ Xavier(w) & \text{weights of non-select dormant neurons} \\ w & \text{otherwise} \end{cases} \tag{B.51}$$

*In Reborn, $g$ map the parameters $\theta$ of the mixing network to $\hat{\theta}$.*

*Proof.*

$$Q_{tot}^{\theta,\phi}(\boldsymbol{\tau},\boldsymbol{u}) \geq Q_{tot}^{\theta,\phi}(\boldsymbol{\tau},\boldsymbol{u}') \quad \exists!k : u_k \neq u_k' \tag{B.52}$$

$$Q_{tot}^{\theta,\phi}(\boldsymbol{\tau},[u_1,...,u_N]) \geq Q_{tot}^{\theta,\phi}(\boldsymbol{\tau},[u_1',...,u_N']) \quad \text{expand } \boldsymbol{u} \tag{B.53}$$

$$f_\theta(Q_1^\phi(\boldsymbol{\tau}_1,u_1),...,Q_N^\phi(\boldsymbol{\tau}_N,u_N)) + V_\theta(\tau) \geq f_\theta(Q_1^\phi(\boldsymbol{\tau}_1,u_1'),...,Q_N^\phi(\boldsymbol{\tau}_N,u_N')) + V_\theta(\tau) \tag{B.54}$$

$$f_\theta(Q_1^\phi(\boldsymbol{\tau}_1,u_1),...,Q_N^\phi(\boldsymbol{\tau}_N,u_N)) \geq f_\theta(Q_1^\phi(\boldsymbol{\tau}_1,u_1'),...,Q_N^\phi(\boldsymbol{\tau}_N,u_N')) \tag{B.55}$$

$$Q_k^\phi(\boldsymbol{\tau}_k,u_k)) \geq Q_k^\phi(\boldsymbol{\tau}_k,u_k')) \quad \exists!k : u_k \neq u_k' \tag{B.56}$$

$$f_{\hat{\theta}}(Q_1^\phi(\boldsymbol{\tau}_1,u_1),...,Q_N^\phi(\boldsymbol{\tau}_N,u_N)) \geq f_{\hat{\theta}}(Q_1^\phi(\boldsymbol{\tau}_1,u_1'),...,Q_N^\phi(\boldsymbol{\tau}_N,u_N')) \tag{B.57}$$

$$f_{\hat{\theta}}(Q_1^\phi(\boldsymbol{\tau}_1,u_1),...,Q_N^\phi(\boldsymbol{\tau}_N,u_N)) + V_{\hat{\theta}}(\tau) \geq f_{\hat{\theta}}(Q_1^\phi(\boldsymbol{\tau}_1,u_1'),...,Q_N^\phi(\boldsymbol{\tau}_N,u_N')) + V_{\hat{\theta}}(\tau) \tag{B.58}$$

$$Q_{tot}^{\hat{\theta},\phi}(\boldsymbol{\tau},\boldsymbol{u}) \geq Q_{tot}^{\hat{\theta},\phi}(\boldsymbol{\tau},\boldsymbol{u}') \tag{B.59}$$

Thus, we show that after the ReBorn process, the learned action preference of DMIX does not change. □

**Corollary 3.** *After the ReBorn Process, the value function of DMIX remain satisfies the IGM principle.*

*Proof.* The proof is the same as the proof for showing after the ReBorn process, QMIX satisfies the IGM principle. It is omitted for brevity. □

Quantile functions (inverse CDF) $\theta$ of a random variable Z is defined as follows.

$$\theta_Z(\alpha) = \inf\{z \in \mathcal{R} : \omega \leq CDF_Z(z)\}, \quad \forall \omega \in [0,1] \tag{B.60}$$

where $CDF_Z(z)$ is the cumulative distribution function of $Z$. We denote $\theta_Z(\omega)$ as $\theta(\omega)$ for simplicity.

**Definition 7** (Conditional Value at Risk(CVaR))**.**

$$CVaR_\alpha(Z) = \mathbb{E}_Z[z|z \leq \theta(\alpha)] \tag{B.61}$$

*where $\alpha$ is the confidence level (risk level), $\theta(\alpha)$ is the quantile function (inverse CDF) defined in (B.60). CVaR is the expectation of values $z$ that are less equal than the $\alpha$-quantile value ($\theta(\alpha)$) of the value distribution.*

**Theorem 6.** *ReBorn with functions $g$ and $h$ satisfies the KI principle for the RMIX [12] value factorization method. RMIX is a risk-sensitive MARL algorithm which consider risk in multi-agent system. Its joint state-action value function is defined as follows.*

$$Q_{tot}^{\theta,\phi}(\boldsymbol{\tau},\boldsymbol{u}) = f_\theta(C_1^\phi(\boldsymbol{\tau}_1,u_1),...,C_N^\phi(\boldsymbol{\tau}_N,u_N)) + V_\theta(\tau) \quad \frac{\partial f}{\partial C_i^\phi} \geq 0 \tag{B.62}$$

$$C_i^\phi(\boldsymbol{\tau}_i,\boldsymbol{u}_i) = CVaR_\alpha[Z_i^\phi(\boldsymbol{\tau}_i,u_i)] \tag{B.63}$$

$$h(w) = w, \quad \forall w \in \phi \quad h \text{ is an identity function} \tag{B.64}$$

$$g(w) = \begin{cases} \beta_i w_x^{in} & \text{input weights of dormant neurons } \boldsymbol{w}_i^{in} \\ \beta_0 w_x^{in} & \text{input weights of over-active neurons } \boldsymbol{w}_x^{in} \\ \frac{1}{\beta_i}\alpha_i w_x^{out} & \text{output weights of dormant neurons } \boldsymbol{w}_i^{out} \\ \frac{1}{\beta_0}\alpha_0 w_x^{out} & \text{output weights of over-active neurons } \boldsymbol{w}_x^{out} \\ \beta_0 b_x & \text{bias of over-active neurons } \boldsymbol{b}_x \\ \beta_i b_x & \text{bias of dormant neurons } \boldsymbol{b}_i \\ Xavier(w) & \text{weights of non-select dormant neurons} \\ w & \text{otherwise} \end{cases} \tag{B.65}$$

*In Reborn, $g$ map the parameters $\theta$ of the mixing network to $\hat{\theta}$.*

*Proof.*

$$Q_{tot}^{\theta,\phi}(\boldsymbol{\tau}, \boldsymbol{u}) \geq Q_{tot}^{\theta,\phi}(\boldsymbol{\tau}, \boldsymbol{u}') \quad \exists! k : u_k \neq u_k' \tag{B.66}$$

$$f_\theta(C_1^\phi(\boldsymbol{\tau}_1, u_1), ..., C_N^\phi(\boldsymbol{\tau}_N, u_N)) + V_\theta(\tau) \geq f_\theta(C_1^\phi(\boldsymbol{\tau}_1, u_1'), ..., C_N^\phi(\boldsymbol{\tau}_N, u_N')) + V_\theta(\tau) \tag{B.67}$$

$$f_\theta(C_1^\phi(\boldsymbol{\tau}_1, u_1), ..., C_N^\phi(\boldsymbol{\tau}_N, u_N)) \geq f_\theta(C_1^\phi(\boldsymbol{\tau}_1, u_1'), ..., C_N^\phi(\boldsymbol{\tau}_N, u_N')) \tag{B.68}$$

$$C_k^\phi(\boldsymbol{\tau}_k, u_k)) \geq C_k^\phi(\boldsymbol{\tau}_k, u_k')) \tag{B.69}$$

$$f_{\hat{\theta}}(C_1^\phi(\boldsymbol{\tau}_1, u_1), ..., C_N^\phi(\boldsymbol{\tau}_N, u_N)) \geq f_{\hat{\theta}}(C_1^\phi(\boldsymbol{\tau}_1, u_1'), ..., C_N^\phi(\boldsymbol{\tau}_N, u_N')) \tag{B.70}$$

$$f_{\hat{\theta}}(C_1^\phi(\boldsymbol{\tau}_1, u_1), ..., C_N^\phi(\boldsymbol{\tau}_N, u_N)) + V_{\hat{\theta}}(\tau) \geq f_{\hat{\theta}}(C_1^\phi(\boldsymbol{\tau}_1, u_1'), ..., C_N^\phi(\boldsymbol{\tau}_N, u_N')) + V_{\hat{\theta}}(\tau) \tag{B.71}$$

$$Q_{tot}^{\hat{\theta},\phi}(\boldsymbol{\tau}, \boldsymbol{u}) \geq Q_{tot}^{\hat{\theta},\phi}(\boldsymbol{\tau}, \boldsymbol{u}') \tag{B.72}$$

Thus, we show that through ReBorn, the learned knowledge about action preference of RMIX does not change. $\square$

**Corollary 4.** *After the ReBorn Process, the value function of RMIX remain satisfies the IGM principle.*

*Proof.* In RMIX, each agent acts greedy according to $C_i^\phi(\boldsymbol{\tau}_i, u_i) = CVaR_\alpha[Z_i^\phi(\boldsymbol{\tau}_i, u_i)]$. It could be viewed as $Q_i^{phi}(\boldsymbol{\tau}_i, u_i)$ in QMIX. By this way, we can prove this Corollary in the same approach as for showing after the ReBorn process, QMIX satisfies the IGM principle. It is omitted for brevity. $\square$

## C  Algorithm

The ReBorn algorithm is described in Algorithm 1.

---

**Algorithm 1** ReBorn

---

**Require:** dormant threshold $\alpha$, over-active threshold $\beta$, reborn interval $T$
 1: Initialize parameters $\theta$ of the mixing network
 2: Initialize parameters $\phi$ of the agent network
 3: Initialize replay buffer $\mathcal{D}$
 4: **for** $e \in \{1, \ldots, m \text{ episodes}\}$ **do**
 5:     Start a new episode;
 6:     **while** episode_is_not_end **do**
 7:         Get the Agent action $a_i$
 8:         Execute $a_i$, obtain global reward $r$ and the next state $s'$
 9:         Update replay buffer $\mathcal{D}$
10:         Sample a batch $\mathcal{D}'$ from replay buffer $\mathcal{D}$
11:         $Loss(\theta, \phi) = (Q(s, a; \theta, \phi) - y_{s,a})^2$
12:         Update $\theta$ and $\phi$ and by $Loss$
13:         **if** $e \bmod T == 0$ **then**
14:            Sample $x$ from replay buffer $\mathcal{D}$
15:            Calculate the $s_i^\ell$ of each neuron in mixing network
16:            Get dormant neurons $dorm_i^\ell$ which $s_i^\ell < \alpha$
17:            Get over-active neurons $over_i^\ell$ which $s_i^\ell > \beta$
18:            **for** each $over_i^\ell$ **do**
19:                Randomly select $K$ $dorm_i^\ell$ that have not been selected before
20:                $over_i^\ell$ assign weights to $K$ $dorm_i^\ell$
21:            **end for**
22:            Reinitialize input weights of unassigned $dorm_i^\ell$
23:            Set output weights of unassigned $dorm_i^\ell$ to 0
24:         **end if**
25:     **end while**
26: **end for**

---

# D  Experimental Details

## D.1  Experimental Setup

We select 4 classical algorithms (QMIX, QPLEX, DMIX, RMIX) with different types to test the generality of ReBorn. QMIX and QPLEX are two well-known value-based MARL value factorization algorithms. DMIX is a distributional MARL value factorization algorithm, while RMIX is a risk-sensitive MARL value factorization algorithm. These four algorithms cover multiple directions in the field of value factorization, demonstrating the strong applicability of ReBorn. Below is the brief descriptions of these algorithms.

Table 1: Baseline value factorization algorithms

| Algorithms | Brief Description |
|---|---|
| QMIX[3] [2] | Learns a mixer of individual utilities with monotonic constraints |
| QPLEX[4] [10] | Learns a mixer of advantage functions and state value functions |
| DMIX[5] [11] | Integrates distributional RL with QMIX |
| RMIX[6] [12] | Integrates risk-sensitive RL with QMIX |

We implement these algorithms based on their open-source repositories to carry out performance analyses, with hyperparameters consistent with those in PyMARL. Our methods are implemented within the PyMARL framework, and each is evaluated using 5 random seeds, with $95\%$ confidence intervals. Specific hyperparameters of different algorithms are listed in Table 2. We conduct experiments on a cluster equipped with multiple NVIDIA GeForce RTX 3090 GPUs.

Table 2: Hyperparameter of different value factorization algorithms

| Hyperparameter | QMIX | QPLEX | DMIX | RMIX |
|---|---|---|---|---|
| Action Selector | epsilon greedy | epsilon greedy | epsilon greedy | epsilon greedy |
| Batch Size | 32 | 32 | 32 | 32 |
| Buffer Size | 5000 | 5000 | 5000 | 5000 |
| Learning Rate | 0.0005 | 0.0005 | 0.0005 | 0.0005 |
| Optimizer | RMSprop | RMSprop | RMSprop | Adam |
| Runner | episode runner | episode runner | episode runner | episode runner |
| Mixing Embed Dimension | 64 | 64 | 64 | 64 |
| Hypernet Embed Dimension | 64 | 64 | 64 | 64 |
| RNN Hidden Dim | 64 | 64 | 64 | 64 |
| Target Update Interval | 200 | 200 | 200 | 200 |
| Discount Factor ($\gamma$) | 0.99 | 0.99 | 0.99 | 0.99 |
| $\alpha$ Dormant Threshold | 0.1 | 0.1 | 0.1 | 0.1 |
| $\beta$ Over-active Threshold | 3 | 3 | 3 | 3 |
| Execution Interval(Step) | 200000 | 200000 | 200000 | 200000 |

In deep RL, ReDo [4] and ReSet [5] are two common mechanisms for improving network's performance through neuron processing. The specific introductions are as follows.

**ReDo.**  ReDo periodically detects dormant neurons within the neural network and resets the input and output weights of these dormant neurons. The input weights are initialized using the Xavier method, while the output weights are set to zero.

**ReSet.**  ReSet periodically resets the parameters of the neural network's final layer using Xavier initialization.

---

[3]https://github.com/oxwhirl/pymarl
[4]https://github.com/wjh720/QPLEX
[5]https://github.com/j3soon/dfac
[6]https://github.com/yetanotherpolicy/rmix

Below is a brief introduction to all the methods used in the experimental section.

Table 3: Methods in Experimental Section

| Algorithms | Brief Description |
|---|---|
| *algorithm* - ReBorn | Apply ReBorn to the Mixing Network of *algorithm* |
| *algorithm* - ReDo | Apply ReDo to the Whole Networks of *algorithm* |
| *algorithm* - ReSet | Apply ReSet to the Whole Networks of *algorithm* |
| *algorithm* - ReBorn (*mechanism*) | Apply *mechanism* to the Mixing Network of *algorithm* |
| *algorithm* - *mechanism* with KI | Apply *mechanism* to the Mixing Network of *algorithm* |
| *algorithm* - *mechanism* w/o KI | Apply *mechanism* to the Whole Networks of *algorithm* |

In Table 3, *algorithm* is QMIX, QPLEX, DMIX, RMIX. *mechanism* is ReBorn, ReDo, ReSet. Specific hyperparameters of different mechanisms are listed as follows.

## D.2 Environment

### D.2.1 Predator-prey

Predator-prey simulates a grid world where multiple agents collaborate to capture preys dispersed throughout the map. At each time step, each agent can choose to *move* or *capture* within its local field of view. A prey is considered captured successfully only when at least two agents around it execute the *capture* action simultaneously. Each successful capture brings a team reward of $+10$, with the goal being to accumulate as much team reward as possible within a limited number of time steps. We develop three distinct environmental configurations: small, middle, large, each featuring different numbers of agents and preys, as well as varying map sizes. Table 4 shows different environmental configurations of Predator-prey in detail.

**Game Rules**

- **Agent Movement**: Agents can move in four directions or stay in place. Movement is restricted by the presence of other agents or preys.

- **Observation and Decision Making**: Each agent observes a 3x3 grid centered around itself, receiving information about nearby agents and preys. Decisions are based on this local observation.

- **Capture Mechanism**: To capture a prey, at least two agents must be adjacent to it and must choose the *capture* action at the same time. Successful capture relies on strategic positioning and synchronized actions among agents.

- **Rewards and Penalties**: Agents receive a positive reward for each prey captured through cooperative action, while individual movement incurs a slight negative time penalty $-0.1$ to encourage efficiency.

- **Episode Termination**: An episode terminates if all preys are captured or after a predefined number of steps, providing a fixed time frame for agents to maximize their collective reward.

| Configuration | Number of Predators | Number of Preys | Map Size | Reward for Capture |
|---|---|---|---|---|
| **Small** | 6 | 12 | 20 x 20 | +10 |
| **Middle** | 12 | 24 | 30 x 30 | +10 |
| **Large** | 18 | 36 | 40 x 40 | +10 |

Table 4: Comparison of Predator-prey Configurations

### D.2.2 StarCraft II Multi-Agent Challenges (SMAC)

The StarCraft Multi-Agent Challenge (SMAC) [13] is a popular benchmark used extensively in the domain of multi-agent reinforcement learning. Built on the StarCraft II game engine, SMAC specializes in micromanagement scenarios where each agent is controlled by an independent agent that must make decisions based on local observations. MARL algorithms coordinate a team of agents to engage in combat against an opposing team managed by the game's built-in AI. The performance of these algorithms is quantitatively evaluated by the test win rate or the test return of the gameplay.

| Name | Difficulty | Ally Units | Enemy Units |
|---|---|---|---|
| 3s_vs_5z | Hard | 3 Stalkers | 5 Zealots |
| 2c_vs_64zg | Hard | 2 Colossi | 64 Zerglings |
| MMM2 | Super Hard | 1 Medivac, 2 Marauders & 7 Marines | 1 Medivac, 3 Marauders & 8 Marines |
| 27m_vs_30m | Super Hard | 27 Marines | 30 Marines |
| 3s5z_vs_3s6z | Super Hard | 3 Stalkers & 5 Zealots | 3 Stalkers & 6 Zealots |
| corridor | Super hard | 6 Zealots | 24 Zerglings |

Table 5: Overview of SMAC scenarios used in the experiment.

Table 5 depicts the overview of SMAC scenarios used in the experiment.

### D.2.3 SMACv2

SMACv2 [14] addresses several critical limitations of SMAC, including the lack of stochasticity and partial observability. Unlike SMAC, SMACv2 features units that are randomly generated and positioned, enhancing stochasticity and significantly increasing the complexity of the scenarios.

| Scenario Name | Number of Allies | Number of Enemies | Unit Types |
|---|---|---|---|
| 10gen_zerg | 10 | 11 | Zergling, Hydralisk, Baneling |
| 10gen_terran | 10 | 11 | Marine, Marauder, Medivac |
| 10gen_protoss | 10 | 11 | Stalker, Zealot, Colossus |

Table 6: Overview of SMACv2 scenarios used in the experiment.

Table 6 depicts the overview of SMACv2 scenarios used in the experiment.

### D.3 Dormant neurons limit the expressive power of Mixing networks

To analyze the impact of the dormant ratio on the expressive power of the mixing network, we consider an illustrative example that requires mixing individual utilities of three agents. For this purpose, we design a simple 2-layer MLP network. The input layer, with a size of 3, receives individual utilities $[Q_i]_{i=1}^3$. The hidden layer contains 4 neurons and uses ReLU as the activation function. The output layer, with a size of 1, produces $Q_{tot}$. The objective is to fit the mixing function $Q_{tot} = 0.5 * Q_1^5 + Q_2^3 + 1.5 * Q_3, Q_i \sim \mathcal{N}(0, 1)$.

We control the dormant ratio by varying the number of dormant neurons in the hidden layer. Figure 2 (a) illustrates the expressive performance of networks with different dormant ratios. *Number = n* indicates that there are $n$ dormant neurons in the hidden layer. We use Mean Squared Error as the loss function. According to the results, an increase in the dormant ratio will lead to reduced expressive power of the mixing network.

### D.4 Experimental Results

### D.4.1 ReBorn can improve the performance of various value factorization algorithms

**Predator-Prey & SMAC**

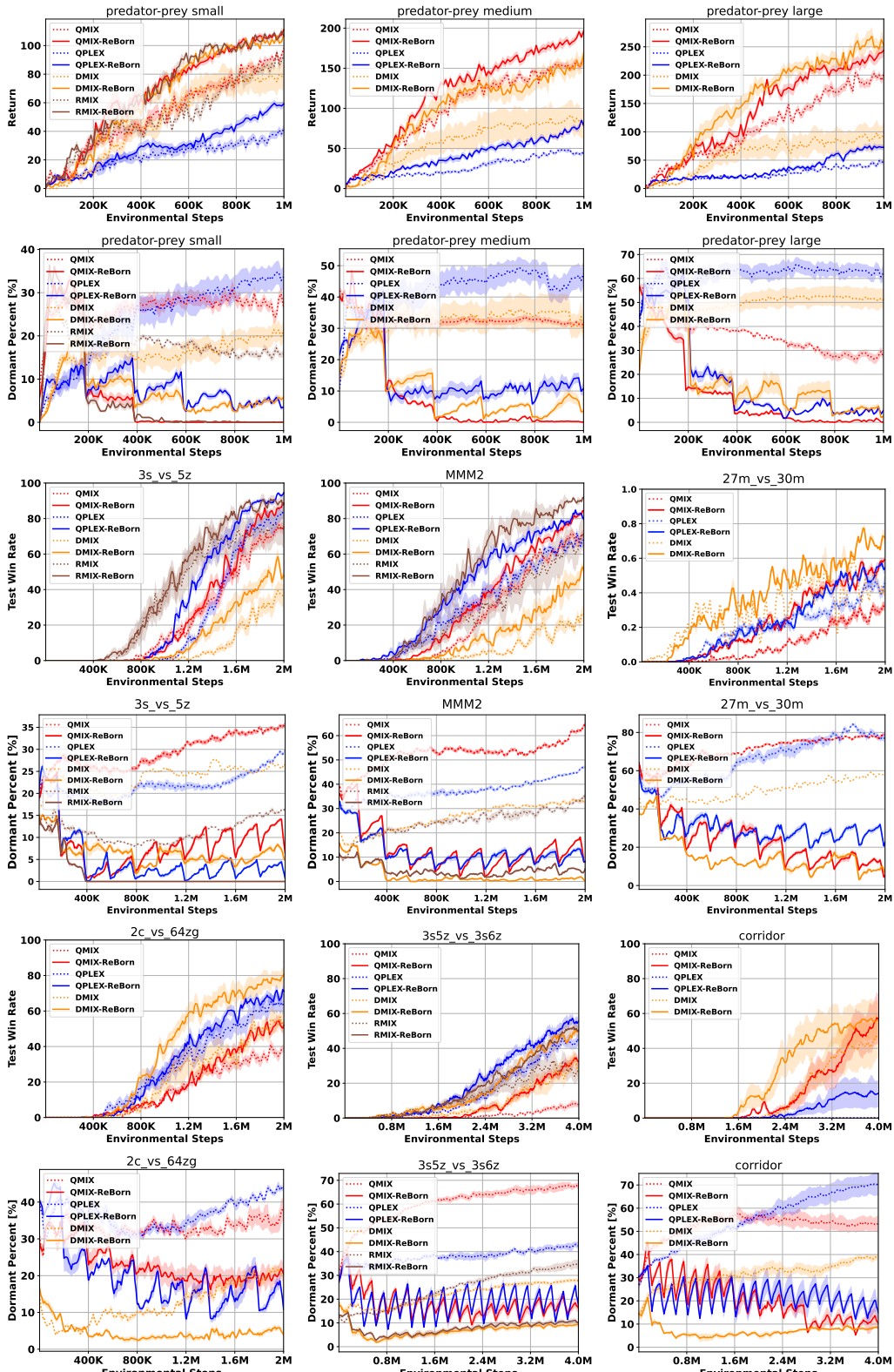

Figure 3: ReBorn can improve the performance of various value factorization algorithms in Predator-Prey and SMAC.

**SMACv2**

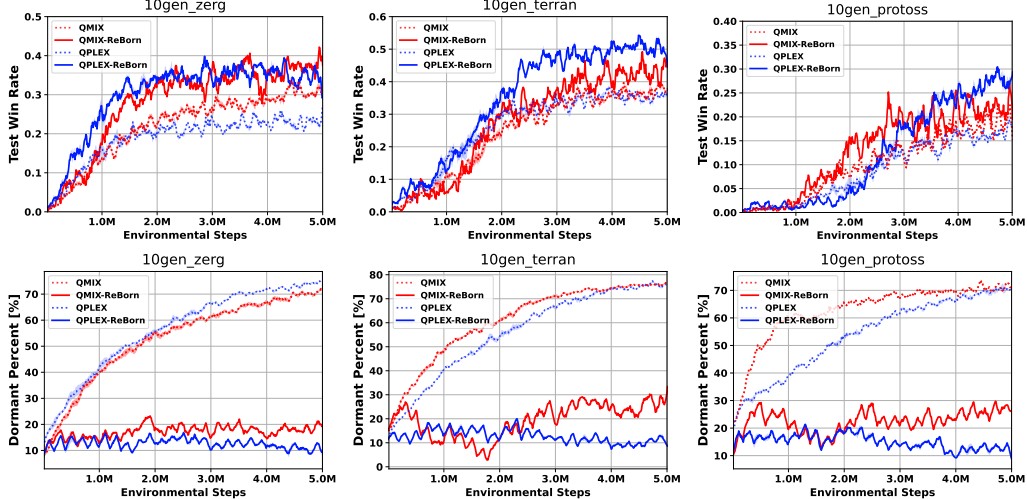

Figure 4: ReBorn can improve the performance of various value factorization algorithms in SMACv2.

### D.4.2 Compare neuron activation values with different methods

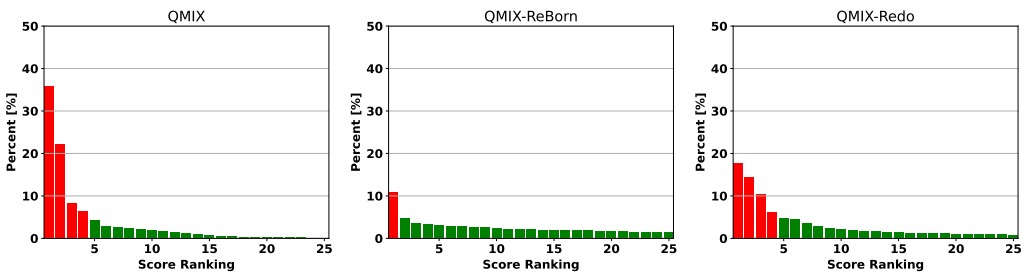

Figure 5: The Normalized Activation Score percentage ranking for top-25 over-active neurons in 27m_vs_30m.

### D.4.3 ReBorn is better than other methods that satisfy the KI principle

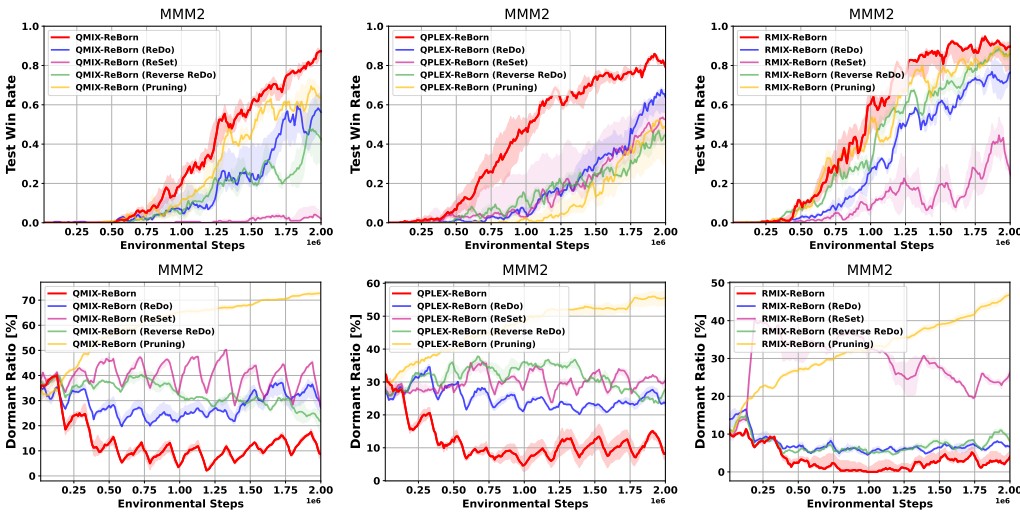

Figure 6: Comparison with other methods that satisfy the KI principle.

## D.4.4 ReBorn is superior to other RL parameter perturbing methods

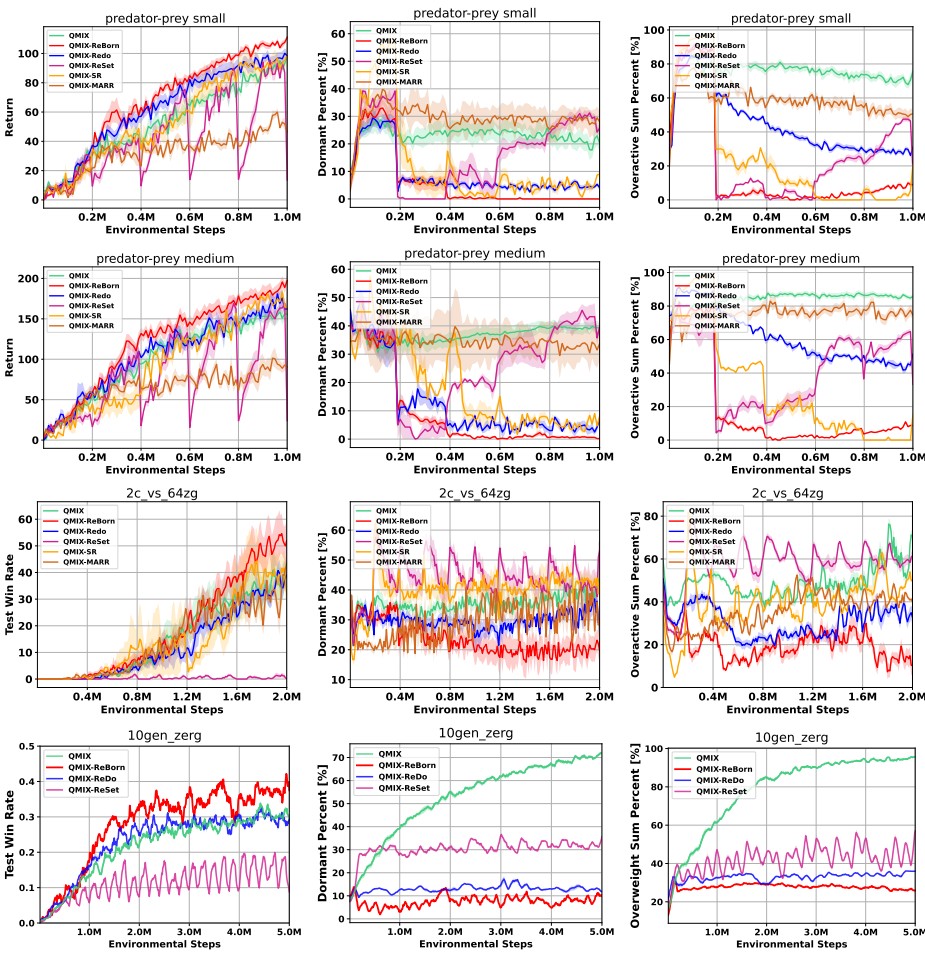

Figure 7: Comparison with Related Methods.

## D.4.5 ReBorn can improve the performance of ResQ

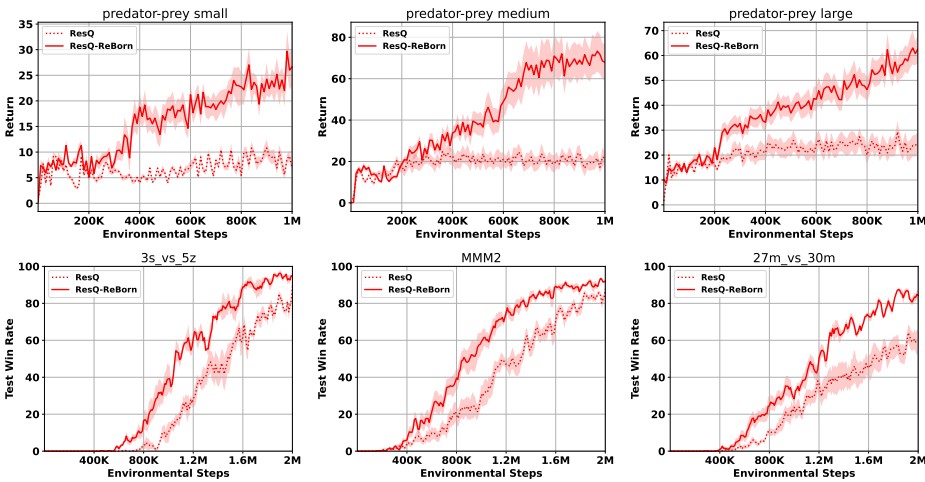

Figure 8: ReBorn can improve the performance of ResQ.

# E   Discussion

## E.1   Societal impact

Our research primarily concentrates on the technical and theoretical aspects of multi-agent reinforcement learning, aiming to enhance the performance of these agents across a variety of tasks. While we do not foresee any direct negative consequences arising from our research, we are committed to maintaining an open dialogue. We highly appreciate and value constructive feedback from the community to ensure our work's contributions are beneficial and ethically sound.

## E.2   Limitations and future work

Although our proposed simple recycling method has achieved good results across various algorithms, there is still room for further improvement. We have defined dormant neurons and over-active neurons in a straightforward manner. However, their identification should not be limited to normalized activation values. More precise identification could be achieved by considering additional factors such as update gradients and output weights.We studied the phenomenon of dormant neurons in discrete multi-agent environments. Future work should explore whether our method can be extended to continuous environments. Regarding different thresholds and recycling periods, setting a threshold too high or recycling too frequently can disrupt the network's normal learning process. Conversely, low thresholds and infrequent recycling can reduce the effectiveness of the recycling process. Therefore, developing adaptive thresholds and recycling mechanisms will be a key focus of future work.

