# OpenReview forum: "The Dormant Neuron Phenomenon in Multi-Agent Reinforcement Learning Value Factorization"
_NeurIPS.cc/2024/Conference — NeurIPS 2024 poster_

### Official Review · Reviewer_WpXH · 2024-07-08

**Soundness:** 3
**Presentation:** 3
**Contribution:** 3
**Rating:** 6
**Confidence:** 4

**Summary:**

The paper proposes ReBorn to reactivate dormant neurons in the mixing network of multiagent value-mixing algorithms. Specifically, ReBorn transfers the weights from overweight neurons to dormant neurons and this method ensures the learned action preferences are not changed after the ReBorn operation. Experiments on SMAC, predator-prey, and SMACv2 demonstrate the effectiveness of the proposed method.

**Strengths:**

1.	The idea is novel and the motivation is well-explained.
2.	Experiments on different scenarios show that ReBorn improves the performance of different baselines.
3.	The authors theoretically prove that ReBorn would not affect the learned action preferences.

**Weaknesses:**

1.	It seems that the ReBorn operation is time-consuming as it needs to compute and manipulate each neuron in the network. If the network is large, this ReBorn operation may become infeasible.
2.	The ReBorn is only validated in the multiagent value-mixing algorithms.

**Questions:**

1.	Could the authors compare the running time of the baselines and their ReBorn variants?
2.	Could ReBorn be applied to other MARL algorithms besides the multiagent value-mixing algorithms such as MAPPO and MADDPG?

**Limitations:**

NA.

---

> ### Author Rebuttal · Authors · 2024-08-06
>
> We thank the reviewer for valuable comments and suggestions, which can help improve the quality of our work. We address the concerns of the reviewer as follows.
>
> >Weakness 1: It seems that the ReBorn operation is time-consuming as it needs to compute and manipulate each neuron in the network. If the network is large, this ReBorn operation may become infeasible.
>
> >Question 1: Could the authors compare the running time of the baselines and their ReBorn variants?
>
>
> The time complexity of the ReBorn operation increases **linearly** with the number of neurons. In the 27m_vs_30m environment, ReBorn is performed 19 times. The training time of the QMIX algorithm increases from 23.8 hours to 24.6 hours. By applying ReBorn, the win rate of QMIX increases from 39% to 51%. The performance benefit (51/39=1.31) bring by ReBorn **outweighs its computational cost** (24.6/23.8=1.03).
>
> We have compared the running time of ReDo and Reset with ReBorn, as well as with two new baselines, SR[1] and MARR[2]. The wall clock time is depicted in the table below. As shown in the table, ReDo, Reset, and ReBorn take almost the same amount of time. Additionally, SR[1] and MARR[2] take more time than ReBorn.
>
> |Map/QMIX|Baseline|ReSet|Redo|ReBorn|SR|MARR|
> |--|--|--|--|--|--|--|
> |3s_vs_5z(2M)|10.2|10.4|10.6|10.9|15.3|11.3|
> |MMM2(2M)|12.3|12.4|13.1|12.9|17.4|13.6|
> |27m_vs_30m(2M)|23.8|24.2|24.7|24.6|29.3|25.3|
> |2c_vs_64zg(2M)|18.3|18.7|19.2|19.3|24.2|19.6|
> |stag_hunt_s(1M)|3.9|3.9|4.2|4.1|6.3|4.8|
> |stag_hunt_m(1M)|7.8|8.0|8.5|8.3|10.3|8.9|
> |stag_hunt_l(1M)|12.7|13.1|13.4|13.6|18.6|14.2|
>
>
> We agree with the reviewer that there could be better ways to reduce the computational overhead of ReBorn. For example, ReBorn could be applied only to a few layers of a neural network rather than the whole neural network. We plan to explore such alternatives in the future.
>
>
>
> > The ReBorn is only validated in the multi-agent value-mixing algorithms.
>
> As it is written in the title of our submission: "The Dormant Neuron Phenomenon in Multi-Agent Reinforcement Learning Value Factorization", we focus on the value factorization (value mixing) algorithms.
>
> We have applied ReBorn to the critic networks of MADDPG and MAPPO, which are not multi-agent value-mixing algorithms. The experimental results are shown in Figure 6 of the response PDF. As it is shown, ReBorn can reduce the dormant ratio of the critic networks of MADDPG  and MAPPO, and improve their performance as well. In future work, ReBorn is expected to be applied in other multi-agent fields.
>
>
>
> References
>
> [1] D'Oro et. al., Sample-Efficient Reinforcement Learning by Breaking the Replay Ratio Barrier, ICLR, 2023
>
> [2] Yang et al., Sample-Efficient Multi-agent Reinforcement Learning with Reset Replay, ICML, 2024

---

> > ### Comment · Reviewer_WpXH · 2024-08-08
> >
> > Thank you for your reply. Most of my concerns are addressed. I have one remaining problem.
> >
> > For the new baselines [1, 2] that were compared, what are the replay ratios for them and ReBorn? It seems that the two methods are working with higher replay ratios larger than 1. Could you compare ReBorn and them at different replay ratio settings such as 1, 5, and 10?

---

> ### Author Response · Authors · 2024-08-11
>
> Dear Reviewer,
>
> We are happy that we have addressed most of your concerns.
>
> For SR[1] and MARR[2], we use their default replay ratios. The replay ratios for SR and MARR are 4 and 1, respectively. The replay ratio for ReBorn is 1.
>
> As per the Reviewer's suggestion, we've conducted a performance evaluation of ReBorn, SR, and MARR for QMIX on the Predata-prey environments at varying replay ratios (1, 5, and 10). Notably, the behavior of ReBorn remains consistent regardless of the replay ratio. However, for SR, the number of reset operations increases with the replay ratio, and for MARR, the number of 'random amplitude scale data augmentation' also increases with the replay ratio.
>
>
>
> For predator-prey small environments, the return and the time (hours) for different algorithms with different replay ratios are depicted as follows. The training time increases with the increase of the replay ratio. We find that ReBorn performs better than SR and MARR when the replay ratio is lower than 5. When the replay ratio is 10, the performance of ReBorn is slightly better than SR and MARR.
>
> |predator-prey small (time)|1|5|10|
> |-|-|-|-|
> |Reborn|4|9|13|
> |SR|4|7|12|
> |MARR|4|9|14|
>
> |predator-prey small (return)|1|5|10|
> |-|-|-|-|
> |Reborn|112|115|114|
> |SR|42|98|113|
> |MARR|51|97|110|
>
>
> For the predator-prey medium environments, the return and the time for different algorithms with different replay ratios are depicted as follows. We find that when the replay ratio is lower than 5, ReBorn performs better than SR and MARR. When the replay ratio is 10, ReBorn performs weaker than SR and MARR.
>
>
>
> |predator-prey medium (time)|1|5|10|
> |-|-|-|-|
> |Reborn|7|13|20|
> |SR|7|11|18|
> |MARR|7|14|22|
>
> |predator-prey medium (return)|1|5|10|
> |-|-|-|-|
> |Reborn|205|203|195|
> |SR|78|162|203|
> |MARR|89|182|208|
>
> Based on our experimental results, it's clear that ReBorn should consider the replay ratio to achieve better performance.
>
>
> References
>
> [1] D'Oro et. al., Sample-Efficient Reinforcement Learning by Breaking the Replay Ratio Barrier, ICLR, 2023
>
> [2] Yang et al., Sample-Efficient Multi-agent Reinforcement Learning with Reset Replay, ICML, 2024

---

> > ### Comment · Reviewer_WpXH · 2024-08-12
> >
> > I appreciate the authors for replying to my question. I have no further questions and I would like to maintain my score. But I recommend including the results of baselines [1, 2] and Reborn with higher replay ratios in SMAC and predator-prey for a fair comparison in the final revision.

---

> > > ### Author Response · Authors · 2024-08-12
> > >
> > > Dear Reviewer,
> > >
> > > We are happy that we have addressed your concerns. We will add the results of baselines[1,2] and Reborn with higher replay ratios in SMAC and predator-prey.
> > >
> > > Best Regards,
> > >
> > > Authors

---

### Official Review · Reviewer_JB6L · 2024-07-09

**Soundness:** 3
**Presentation:** 2
**Contribution:** 3
**Rating:** 5
**Confidence:** 4

**Summary:**

This paper introduces a novel approach for tackling the plasticity loss and securing sample efficiency in multi agent reinforcement learning (MARL). Different from the methods in single agent RL, this paper also shows the importance of not violating the knowledge invariant (KI) principle in MARL. With careful analysis and extensive experiments, the proposed method, ReBorn, outperforms other baselines such as ReDo or Reset. The authors said that the baselines, ReDo and Reset, violate the KI principle, thus they are not appropriate to be applied to MARL setting. However, the proposed method, ReBorn, can satisfy the KI principle, leading to achieve remarkable performance.

**Strengths:**

- One of the most important part in this paper, the KI principle, is well defined, and can represent the constraints in MARL setting. Due to the difficulty of satisfying this principle, applying previous methods is not straightforward. Nevertheless, the proposed method, ReBorn, does not violate this principle, and outperforms the baselines.

- Scaling the input and output weights connected to overweight neurons is a novel approach for resolving the plasticity loss problem in Rl community. This paper pointed out the drawback of ReDo in MARL scenario, and proposed a novel approach.

**Weaknesses:**

- The motivation behind scaling the weights in ReBorn is not clear. Though scaling the weights to ease the strength of the overweight neurons is straightforward, is it only way to reduce the number of overweight neurons? I'm quite confusing why we should use the scaling approach in ReBorn.

- I think there is a missing baseline [1]. This approach can also effectively resolve the plasticity loss problem, and achieves high sample efficiency in singe agent RL. Although this approach violates the KI principle, with high replay ratio, it may achieve good performance in MARL setting. Similar to [1], a recently proposed method [2]  show that resetting the network and training the network with high replay ratio in MARL setting can outperform the baselines. I know it is difficult to  compare ReBorn and the method in [2] at submission phase, but I wonder the method in [2] violating the KI principle is truly failed to overcome the plasticity loss.


[1] D'Oro et. al. ,Sample-Efficient Reinforcement Learning by Breaking the Replay Ratio Barrier, ICLR, 2023

[2] Yang et. al., Sample-Efficient Multiagent Reinforcement Learning with Reset Replay, ICML, 2024

**Questions:**

Already mentioned in the Weakness section.

---

> ### Author Rebuttal · Authors · 2024-08-06
>
> We thank the reviewer for your time and effort in reviewing this work. Your suggestions are valuable. We have compared our method with the two approaches mentioned by the reviewer. We address the concerns of the reviewer as follows.
>
> >The motivation behind scaling the weights in ReBorn is not clear. Though scaling the weights to ease the strength of the overweight neurons is straightforward, is it only way to reduce the number of overweight neurons? I'm quite confusing why we should use the scaling approach in ReBorn.
>
> We agree with the reviewer that there could be multiple ways to ease the strength of the overweight neurons. We did explore another way of distributing the weights, which set $\alpha_i$ = $1/(M+1)$, and $\beta$ = 1. As shown in Figure 4 of the response PDF, such an average weight-sharing method performs inferior to ReBorn; its performance is weaker than the weight scaling approach. We would like to explore the method used by [2], which uses a Shrink & Perturb strategy.
>
> >I think there is a missing baseline [1]. This approach can also effectively resolve the plasticity loss problem and achieve high sample efficiency in single-agent RL. Although this approach violates the KI principle, with a high replay ratio, it may achieve good performance in the MARL setting. Similar to [1], a recently proposed method [2] shows that resetting the network and training the network with a high replay ratio in the MARL setting can outperform the baselines. I know it is difficult to compare ReBorn and the method in [2] at the submission phase, but I wonder if the method in [2] violating the KI principle truly failed to overcome the plasticity loss.
>
> We will add the following discussion to the related work section. *[1] increases replay ratio and resets all the parameters of networks based on the number of updates. For MARL, [2] periodically resets the parameters of agent networks and uses data augmentation to further increase the replay ratio.*
>
>
> We have evaluated the performance of [1] and [2] in four more environments. The results are depicted in Figure 2 of the response PDF. As shown in Figure 2, these two methods perform inferior to ReBorn. We will add these experimental results to our work.
>
> We agree with the reviewer that the KI principle offers a perspective on the plasticity loss problem. There could be more perspectives on this issue.
>
> References
>
> [1] D'Oro et. al., Sample-Efficient Reinforcement Learning by Breaking the Replay Ratio Barrier, ICLR, 2023
>
> [2] Yang et al., Sample-Efficient Multiagent Reinforcement Learning with Reset Replay, ICML, 2024

---

> ### Author Response · Authors · 2024-08-13
>
> Dear Reviewer,
>
> Thanks for your time and effort in reviewing our work. We greatly appreciate your insightful and detailed feedback. We have carefully addressed the concerns in the rebuttal. Please let us know if any aspects remain unclear, and we are happy to provide further clarification.
>
> Best regards,
>
> Authors

---

### Official Review · Reviewer_hhg9 · 2024-07-12

**Soundness:** 4
**Presentation:** 2
**Contribution:** 4
**Rating:** 7
**Confidence:** 3

**Summary:**

The paper explores the dormant neuron phenomenon, an active research topic in RL, in the context of MARL value factorization. It describes how dormant neurons manifest mostly in the mixing network, details their effect on performance and connects them to an opposite but correlated phenomenon, overweight neurons. The paper proposes a method to counter the dormant neuron phenomenon by redistributing the weights of overweight neurons among dormant neurons. In MARL value factorization, the proposed method is demonstrated to outperform general RL methods for dealing with dormant neurons.

**Strengths:**

The paper text is easy to read and understand, and the problem the paper investigates is important and topical. Evaluation shows that, on two environments, the proposed method (ReBorn) is clearly better than general RL methods used to solve this problem. The discussion on the existence of overweight neurons is also significant for the field outside of the specific problem setting.

**Weaknesses:**

The figures and figure captions are often hard to understand. Some plots are hard to interpret solely from the plot and caption text, e.g. to understand what 'Number' means in Figure 2a, one must look for explanations in the text body. Another example is Figure 4, which the reader is referred to for a depiction of the main method of the paper, but contains a total of 8 words in legends and caption combined. In contrast, the figures themselves are visually clear to understand.
This problem can be easily fixed by expanding some of the captions and using more expressive wording in figure legends and titles. I am willing to raise my score if the figures are presented in a more readable manner.

**Questions:**

Suggestion:
Evaluation on more environments could strengthen the paper's claim of superiority over other methods. However, I find the current evaluation experiments sufficient to make that claim.

**Limitations:**

Limitation are not discussed in the main section, only in the appendix.

---

> ### Author Rebuttal · Authors · 2024-08-06
>
> We would like to thank the reviewer's time and effort to review this paper. We will improve the quality of the figures and add more experimental results to our work.
>
> >The figures and figure captions are often hard to understand.
>
> Thanks for your suggestion on Figure 2a and Figure 4. We have improved the presentation of these two figures. They are presented in Figures 1 and 3 in the response PDF. Additionally, we have improved the readability of Figure 2(b), Figure 2(c), Figure 3(a-c), Figure 5(a-c), Figure 6 (a-c), Figure 7 (a-c), and Figure 8 (a-c) as follows.
>
> For Figure 2a, we have changed its vertical label from "Loss" to "MSE Loss for a Mixing Network" and changed its legend from "Number" to "Number of Dormant Neurons". Further, we have also updated the caption of the Figure from "Dormant neurons hurt mixing network expressivity" to "The MSE Loss for fitting a simple Mixing Network increases with an increasing number of Dormant Neurons. It indicates that dormant neurons hurt mixing network expressivity."
>
> For Figure 4, we have updated the caption of the Figure from "ReBorn Method" to "The procedure of ReBorn neurons. The weights of overweight neurons are distributed to $M$ randomly picked dormant neurons. $w_x^{in}$, $w_x^{out}$, and $b$ are the input weights, output weight, and bias for an overweight neuron. $w_i^{in}$, $w_i^{out}$ and $b_i$ are the input weights, output weight, and bias for dormant neuron $i$. After Reborn, the input weights and bias for a dormant neuron $i$ becomes $\beta_i\alpha_iw_x^{in}$, and $\beta_ib_x$, where $0\leq \alpha_i \leq 1, and \sum_{i=0}^M\alpha_i=1$, $\beta_i$ is a random number among [0.5, 1.5]."
>
>
> For Figure 2(b), we have relocated the legends from the top-left to the bottom-right and changed the legend from "Interval" to "Update Interval."
>
> For Figure 2(c), we have changed its vertical label from "Percent" to "NAS Percentage", and changed its horizontal label from "Score ranking" to "NAS ranking for top-25 neurons". We have updated its caption from "Overweight neurons in the QMIX mixing network" to "The Normalized Activation Score (NAS) percentage ranking for top-25 overweight neurons in the QMIX mixing network."
>
> For Figure 3, we have updated the caption to "Overweight neurons in QMIX mixing networks: (a) The percentage contribution of the number of dormant neurons (depicted as Dormant), the number of overweight neurons (depicted as Overweight-Number), the sum of NAS (depicted as Overweight-Sum) for overweight neurons over time. (b) Overlap coefficients for Dormant/Overweight neurons between the current iteration and previous iterations. (c) Percentage of dormant neurons that re-enter dormancy after ReDo within different time steps."
>
> For Figure 5 (a-c), we have changed the caption from "ReBorn can improve the performance of various value factorization algorithms" to "ReBorn can improve the performance of various value factorization algorithms: (a-b) the test win rate for the 3s5z\_vs\_3s6z and the MMM2 environments, (c) the return for predator-prey small environment, (d-e) the dormant percent for the the 3s5z\_vs\_3s6z, the MMM2, and the predator-prey small environment."
>
> For Figure 6 (a-c), we have updated the caption from "Comparison with other Parameter Perturbing Methods." to "Comparison with other Parameter Perturbing Methods: (a-c) The test win rate, the dormant percentage and the percentage of the sum of normalized activation score (NAS) for the MMM2 environment. (d-f) The test win rate, the dormant percentage, and the percentage of the sum of NAS for the 27m\_vs\_30m environment."
>
> For Figure 7 (a-c), we have changed the caption from "Importance of satisfying the KI Principle" to "Importance of satisfying the KI Principle for (a) QMIX, (b) QPLEX, and (C) RMIX. A variant of ReBorn without satisfying the KI Principle is depicted as Reborn w/o KI."
>
> For Figure 8 (a-c), we have changed the caption from "Comparison with other methods that satisfy the KI principle" to "Comparison with other methods that satisfy the KI principle for (a) QMIX, (b) QPLEX, and (C) RMIX."
>
>
> >Evaluation on more environments could strengthen the paper's claim of superiority over other methods. However, I find the current evaluation experiments sufficient to make that claim.
>
> We have evaluated ReBorn against ReDo and Reset in four more environments. The experimental results are shown in Figure 2 in the response PDF. The results demonstrate that ReBorn performs better than these two methods.
>
>
> >Limitation are not discussed in the main section, only in the appendix.
>
> Thank you for your valuable suggestion. We will incorporate the discussion of the limitations in the main section.

---

> > ### Comment · Reviewer_hhg9 · 2024-08-11
> >
> > Thank you for your response. Since you have addressed my concerns, I will raise my score to 7.

---

### Official Review · Reviewer_KvCg · 2024-07-14

**Soundness:** 2
**Presentation:** 3
**Contribution:** 3
**Rating:** 5
**Confidence:** 4

**Summary:**

This work proposes a new method for resetting dormant neurons in multi-agent RL settings. It replicates previous observations on unit dormancy in deep RL in the multi-agent regime, finding that inactive neurons are correlated with reduced ability to improve performance. The proposed method differs from ReDO, which resets dormant neurons' incoming weights to that of a random initialization, by distributing the weights of highly active "overweight" neurons among those identified as dormant during reset steps so as to avoid changing the output of the network.

**Strengths:**

- The paper identifies additional factors, beyond those present in single-agent deep RL, which impact the dormancy rate in a neural network, in particular showing that in QMIX and QPLEX algorithms, the *mixing* network is most vulnerable to dormancy. It also shows that increasing the number of agents, which presumably also increases the degree of nonstationarity in the problem, also increases the number of dormant units.
- The knowledge invariant principle, while not entirely novel (see, e.g. the approach of Nikishin et al. which is precisely motivated by the desire to avoid changing the network's outputs after a reset), is clearly relevant for multi-agent settings, where changing many agents' policies at once introduces greater instability than the (potentially beneficial) exploratory interpretation of parameter perturbations in single-agent algorithms.
- The observation that the accumulation of dormant neurons is accompanied by accumulation of 'overweight' neurons is sensible and points to an interesting alternative approach to maintaining plasticity: rather than focusing on resetting units which are not receiving gradients, it may be beneficial to re-distribute utility across the units in a layer. The proposed solution makes a lot of sense from this framing.

**Weaknesses:**

I have two main concerns with this paper which prevent me from confidently recommending its acceptance: the first is the validity of the theoretical results, and the second is signficance of the empirical results. I list more minor concerns later, which may benefit the paper but which will not affect my decision.

- Major: while it is true that a parameter perturbation which changes the rankings of global actions can cause a function which previously satisfied the IGM principle to violate it, such a perturbation would also violate the assumption on the monotonicity of the global value w.r.t. agent utilities which is baked into methods like QMIX. Indeed, the proof of Theorem 1 does not provide a concrete example of a perturbation to the network parameters which violate the KI principle and violate the IGM principle, as such an example which satisfies [B.5] would require non-monotonicity of the mixing function. Since the algorithms studied in the paper do use monotone mixing functions, Theorem 1 does not seem relevant.

- Major: I could not find anything in the proof of Theorem 2 which depends on the particular form of the ReBorn update. Instead it seems that Theorem 2 depends on the monotonicity of the QMIX mixing network (b.10), rather than any particular property of the perturbation to $\hat{\theta}$. This further reinforces the above concern that the proposed failing of parameter perturbation methods in the general case of Theorem 1 is in fact only a product of relaxing the assumption on the monotonicity of the mixing network.

- Major: In many instances, using ReDo appears to hurt performance compared to naive QMIX, despite reducing the dormancy percentage. Further, it appears that in some environments, applying ReDo does not change the number of dormant neurons. This causes me to wonder if the ReDo method has been appropriately tuned for the domains it is being applied to.

- Minor: to me, a better name for the so-called 'overweight' neurons would be 'over-active', since it's not clear that the cause of their disproportionate activity is due to the magnitude of their weights or to the alignment of their incoming weights with the previous layer's features. This also makes it more clear their relationship with dormant neurons.

- Minor: there are a few grammatical issues in the paper that would benefit from review. For example, the word 'albeit' is typically used at the start of a phrase, and not a clause as is often done in the paper, where it can generally be replaced with 'although'.

**Questions:**

1. Can the authors comment on how they tuned ReDo and give some insight into why it seems to not have any effect on the number of dormant units in Figure 6?
2. Please address my questions regarding theorems 1 and 2. Are there instances where even with a monotone mixing network, a parameter-perturbing method will fail to satisfy IGM?

**Limitations:**

Yes

---

> ### Author Rebuttal · Authors · 2024-08-06
>
> Thanks for your valuable comments, we will improve our work based on your suggestions. We address your concerns as follows.
>
> >The knowledge invariant principle, while not entirely novel (see, e.g., the approach of Nikishin et al., which is precisely motivated by the desire to avoid changing the network's outputs after a reset)
>
> The approach of Nikishin et al. is the Reset method used for comparison. It aims to preserve the experience within the replay buffer and the un-reset layers. However, it does not try to avoid changing the output of network.
>
>
> >Major 1: while it is true that a parameter perturbation which changes the rankings of global actions, ... not provide a concrete example of a perturbation to the network parameters which violate the KI principle and violate the IGM principle, ... require non-monotonicity  ... Theorem 1 does not seem relevant.
>
>
> Here, we present an example that violates the KI and IGM principles. Assuming we have trained a QMIX-like value factorization method $f$ that implements $Q_{tot}=\sum_{i=1}^Nk_i\times Q_i$, where $k_i \ge 0$. It is implemented by a two layer-neural network parameterized by $k_i$. In the last layer, there is only one neuron (output neuron). For each neuron $i$ in the first layer, its input is individual utility $Q_i$, and the weight connected to the output neuron is $k_i \ge 0$. The activation function of the output neuron is an identity function. Clearly, $f$ is a monotonic increasing network, satisfying the IGM principle. If a parameter perturbing method changes all the weight $k_i$ to be smaller than zero, then $f$ does not satisfy the IGM principle, and the KI principle is violated.
>
> We agree with the reviewer that *[B.5] would require non-monotonicity of the mixing function*. We have shown that a perturbing method could change a mixing function from monotonic increasing to monotonic decreasing.
>
>
> Besides monotonic increasing mixing functions (e.g., QMIX and RMIX), ReBorn supports both DMIX and QPLEX. DMIX has a **non-monotonic increasing** mixer for value distributions and QPLEX can model **non-monotonic** relationships. We implement ReBorn on the hypernet for QMIX, on the QMIX part of DMIX, and advantage part of QPLEX to ensure that we do not change their functional relationships. We will make this clearer.
>
>
> >Questions 2: Are there instances where even with a monotone mixing network, a parameter-perturbing method will fail to satisfy IGM?
>
> If a parameter-perturbing method changes the functional property of the mixing network, it will fail to satisfy IGM. If the functional property is preserved, it will not fail to satisfy IGM.
>
> >Major 2: I could not find anything in the proof of Theorem 2 which depends on the particular form of the ReBorn update. ... only a product of relaxing the assumption on the monotonicity of the mixing network.
>
> The proof of Theorem 2 does depend on the implementation of ReBorn on QMIX. We will make it clearer in the paper. We apply ReBorn to the neurons of hypernetworks used in QMIX. Hypernetworks are used to generate non-negative weights of the mixing network. A hypernetwork takes $\tau$ as input and outputs the weights of a layer of the QMIX mixing network. A hypernetwork consists of a single linear layer, followed by an activation function, which ensures that the mixing weights are non-negative after ReBorn.
>
> After applying ReBorn, the monotonic increasing property of QMIX does not change. We do not perturb the agent network, as we find that the dormant ratio is low in the agent network. In Sec 6.4.1, we apply ReBorn both to the mixing network and the agent network. This leads to a violation of the KI principle. As shown in Fig. 7, depicted as ReBorn w/o KI, applying it only to the mixing network leads to better performance.
>
>
>
> >Major 3: In many instances, using ReDo appears to hurt performance compared to naive QMIX, despite reducing the dormancy percentage ... applying ReDo does not ... dormant neurons ...  if the ReDo method has been appropriately tuned ...
>
> >Question 1: Can the authors comment on how they tuned ReDo and give some insight into why it seems to not have any effect on the number of dormant units in Figure 6?
>
> **For Figure 6, ReDo can reduce dormant ratios.** The middle of Figure 6 (b, e) depicts the dormant ratios for different methods. The results for ReDo (QMIX-ReDo) are depicted as a blue curve, showing that it can reduce the dormant ratio for the QMIX algorithm (depicted as a green curve).
>
> ReDo can **reduce the dormant ratio** for all the cases shown in this work. Please refer to all the figures (Fig. 6(b, f), Appendix Fig. 5 middle, Appendix Fig. 7 bottom row) that depict the dormant ratio for Redo. Regarding performance, ReDo hurts the performance of QMIX on MMM2, but it increases its performance for 27m-vs-30m, which has a higher dormant ratio than MMM2.
>
> For a fair comparison, we performed parameter turning on ReDo for 2 SMAC environments (MMM2 and 5m vs 6m) based on ReDo's default parameters, which perform well for 47 single agent environments. We explored two parameter re-initialization methods (Xavier and Kaiming) and two dormant thresholds (0.025 and 0.1). The results are depicted in Figure 5 of the response PDF. Different configurations perform similarly. In the end, we choose the default parameters of ReDo.
>
> As we discussed in Sec. 4, the presence of over-active neurons impacts the existence of dormant neurons. Therefore, ReDo's low efficiency in reducing dormant neurons might be due to its inability to handle over-active neurons. This is a factor contributing to the effectiveness of ReBorn, which addresses both dormant and over-active neurons.
>
>
> >Minor: a better name for the so-called 'overweight' neurons would be 'over-active'.
>
> We will change the word "overweight" to "over-active".
>
> >Minor: there are a few grammatical issues in the paper ...
>
> We will change the 'albeit' in a clause to 'although'. We will check the paper carefully to improve its overall quality.

---

> ### Comment · Reviewer_KvCg · 2024-08-09
>
> Thanks to the authors for their response.
>
> - **Nikishin et al. citation:** Apologies for the ambiguity, Nikishin et al. reference I referred to was not the primacy bias work but rather the later paper on plasticity injection [1], which proposes to freeze the network parameters and initialize a new, trainable network whose output is added to that of the frozen network in order to increase trainability.
> - **Monotonicity:** I appreciate the provided example, however this example also seems to depend on an unconstrained mixing network in which there exist parameters under which the output is non-monotonic in the inputs. If I understood correctly, however, QMIX and related methods explicitly constrain their mixing networks to be monotonic under all possible parameterizations. The major aspect of my concern which is still unclear after the authors' response is what property of ReBorn ensures that it will satisfy the KI / IGM principle in a situation where ReDo does not. Providing a concrete worked example of such a case would help to address this concern.
> - Could the authors clarify what they mean by “functional property” in their response?
> - **ReDo baseline:** While I agree with the authors that ReDo is reducing the dormant neuron ratio relative to the baseline, the specific feature that drew my attention in Figure 6 was the relatively flat slope of the dormant neuron curve in the second half of training in the bottom middle panel, where it seems like each time the ReDo algorithm is called later in training it does not even transiently reduce the number of dormant neurons. This result would benefit from a brief discussion of why applying ReDo is not even temporarily reducing the number of dormant neurons.
>
> [1] Nikishin, Evgenii, et al. "Deep reinforcement learning with plasticity injection." Advances in Neural Information Processing Systems 36 (2024).

---

> ### Author Response · Authors · 2024-08-12
>
> Dear Reviewer,
>
> Thanks for your reply. We address your concerns as follows.
>
> >Nikishin et al. citation: Apologies for the ambiguity, Nikishin et al. reference I referred to was not the primacy bias work but rather the later paper on plasticity injection [1], which proposes to freeze the network parameters and initialize a new, trainable network whose output is added to that of the frozen network in order to increase trainability.
>
> Thanks for the clarification. Plasticity injection (PI) is an interesting work. It aims to increase the plasticity without changing the neural network output by adding new parameters. Unlike PI, our work does not introduce new parameters. We will discuss it in the related work section.
>
> We have applied PI in the agent network (Agent+PI), in the mixing network (Mixer+PI), and in both the agent and mixing network (Agent+Mixer+PI) for QMIX in the predator-prey small, predator-prey medium, MMM2 environments. For these experiments, we use the default hyperparameters of PI. The experimental results are listed as follows. The return is reported for predator-prey, and for MMM2, the win rate is reported. The experimental results show that PI performs inferior to ReBorn.
>
>
>
> |QMIX|predator-prey small|predator-prey medium|MMM2|
> |-|-|-|-|
> |Agent+PI|106|173|46|
> |Mixer+PI|67|114|64|
> |Agent+Mixer+PI|73|125|44|
> |Reborn|112|205|83|
>
>
> >Monotonicity: I appreciate the provided example, ... The major aspect of my concern which is still unclear after the authors' response is what property of ReBorn ensures that it will satisfy the KI / IGM principle in a situation where ReDo does not. Providing a concrete worked example of such a case would help to address this concern.
>
> Thanks for your question; we will make our writing clearer. We will discuss different variants of ReDo more clearly, and include related experiments.
>
> In single-agent RL, ReDo perturbs the parameters of the agent network. For MARL, ReDo perturbs the agent and mixing networks through functions $h$ and $g$, respectively. $h$ and $g$ are the same; they both use Xavier initialization to re-initialize the dormant neurons. As the parameters of the agent network are perturbed through $h$, **the local ranking of actions could be changed, and so does the global ranking**; thus, we state that it does not guarantee to satisfy the KI principle.
>
> In Section 6.4.2, we replace $h$ used by ReDo with an identity function $h(\theta) =\theta$ used by ReBorn, which **does not perturb agent network parameters**. The new method is named ReBorn(ReDo). We have stated in Section 6.4.2 that ReBorn(ReDo) **satisfies the KI principle**. The experimental results show that ReBorn(ReDo) performs inferior to Reborn (Figure 8). Additionally, we apply ReDo to the agent network only by using the function $g$ for the agent network and the identity function $h$ for the mixing network. In the following table, we show the result for using the $g$ function only for the agent network (Agent-ReDo), for the mixing network (Mixer-Redo), for both the agent network and the mixer network (ReDo), and ReBorn. Mixer-Redo is called ReBorn(ReDo) in Figure 8. For predator-prey small and predator-prey medium, the return is reported. For MMM2, the win rate is reported.
>
>
> |QMIX|predator-prey small|predator-prey medium|MMM2|
> |-|-|-|-|
> |Agent-ReDo|88|165|46|
> |Mixer-ReDo|105|184|59|
> |ReDo|92|177|55|
> |ReBorn|112|205|83|
>
>
> For Agent-ReDo and Mixer-ReDo, the identity function $h$ is applied to the mixing and agent networks, respectively. We find that by using $g$ only to the agent network (Agent-ReDo), its performance is poor due to the violation of KI and the existence of dormant neurons, mainly in the mixing network. Mixer-ReDo performs better than Agent-ReDo and ReDo, thanks to the satisfaction of the KI principle. However, all these variants of ReDo perform inferior to Reborn.
>
>
> The $h$ and $g$ functions used by ReBorn are different from ReDo. As it is written in lines 226-227, $h$ is an identity function used for the agent network, and $g$ is applied for the mixing network through weight-sharing among over-active and dormant neurons. ReBorn perturbs the parameters of the mixing network rather than the agent network, thanks to the discovery that there exist few dormant neurons in the agent network. For ReBorn, as the agent network is not perturbed, **the local ranking of action is not changed, so does the global ranking of actions for a monotonic mixer**. We have stated in line 287 and 289 that, *if we applied the ReBorn function $g$ to the agent network, the KI principle will be violated*, and lead to poor performance in Section 6.4.1 (Figure 7).

---

> ### Author Response · Authors · 2024-08-12
>
> >Could the authors clarify what they mean by “functional property” in their response?
>
> Functional property means the monotonicity or the constraints for a value factorization method to satisfy the IGM principle.
>
> For some mixers (e.g., QTRAN), although they follow the IGM principle, they may fail to satisfy it due to parameter perturbation. For QTRAN, the constraints (Theorem 1, Formula 4b in [1]) that guarantee the IGM principle are implemented through a mean-square loss, not through neural network features (such as absolute activation function). Through parameter perturbing, the IGM principle for QTRAN could be violated.
>
>
> >ReDo baseline: While I agree with the authors that ReDo is reducing the dormant neuron ratio relative to the baseline, the specific feature that drew my attention in Figure 6 was the relatively flat slope of the dormant neuron curve in the second half of training in the bottom middle panel, where it seems like each time the ReDo algorithm is called later in training it does not even transiently reduce the number of dormant neurons. This result would benefit from a brief discussion of why applying ReDo is not even temporarily reducing the number of dormant neurons.
>
> ReDo is called every 0.2 million (roundly) steps, and we report the dormant ratio every 10,000 (roundly) steps. **Most of the time when ReDo is performed, the dormant ratio before ReDo is reported, not the ratio just after ReDo**. The middle top of Figure 6 depicts the ratio for MMM2. The ratio curve does experience a temporary drop after ReDo is called.
>
> The middle bottom of Figure 6 depicts the ratio for the 27m vs 30m scenario, which has more agents (27 agents) than MMM2 (12 agents). It is more non-stationary with a higher dormant ratio than MMM2. **When ReDo is called, the dormant ratio does drop immediately, but it increases quickly to its previous value within the next time window when the ratio is reported.** Thus, a temporary drop in the ratio is not observed. The following table shows the dormant ratio with respect to time steps for one run of ReDo on 27m vs 30m. For this table, "Report" indicates the dormant ratio is reported in a figure; "ReDo" indicates that ReDo is performed. For step 1796103, before ReDo is called, the dormant ratio is 30. At step 1796226, just after ReDo, the dormant ratio drops from 30 to 9. **This indicates that ReDo does work as it is designed.** Such ratio quickly increases from 9 to 29 at step 1806262, when the ratio is reported in a figure. Thus, such a temporary drop is not observed.
>
> |Time step|Dormant ratio|Action|
> |-|-|-|
> |1786065|31|Report|
> |...|...|...|
> |1796103|30|Report|
> |1796226|9|ReDo|
> |1796300|10||
> |1796480|16||
> |1796518|15||
> |...|...|...|
> |1801093|21||
> |...|...|...|
> |1806262|29|Report|
>
>
> Reference
>
> [1] Son et al. QTRAN: Learning to Factorize with Transformation for Cooperative Multi-Agent Reinforcement learning, ICML 2019

---

### Author Rebuttal · Authors · 2024-08-06

We sincerely thank all reviewers for their insightful comments and valuable feedback. The reviewers acknowledge our work as novel (KvCg, JB6L, WpXH), important (hhg9, JB6L), significant outside the specific problem setting (hhg9), theoretical contributions (KvCg, JB6L, WpXH), good empirical results (hhg9, JB6L, WpXH). We will incorporate the suggestions and address the concerns in the new version of our work. We have conducted 56 additional experiments to address the comments, and their results are included in the response PDF. We describe the figures as follows.

**Experimental Results**

Fig. 1: We have re-designed the figure and improved its caption to make it more readable.

Fig. 2: Comparing ReBorn with more new baseline[1][2], Reset, ReDo. The experimental results show that ReBorn performs better than them.

Fig. 3: The MSE Loss for fitting a simple Mixing Network increases with an increasing number of Dormant Neurons. We have improved the legend and the caption of the figure to enhance readability.

Fig. 4: Replacing the weight scaling strategy used in ReBorn with an average weight sharing strategy. The experimental results indicate that the weight scaling strategy performs better.

Fig. 5: Hyper-parameter tuning for the ReDo method. These Hyper-parameters exhibit similar performance.

Fig. 6: Applying ReBorn to the critic network in MADDPG and MAPPO. ReBorn can reduce the dormant ratio of the critic network and improve its performance as well.

**References**

[1] D'Oro et al., Sample-Efficient Reinforcement Learning by Breaking the Replay Ratio Barrier, ICLR, 2023

[2] Yang et al., Sample-Efficient Multiagent Reinforcement Learning with Reset Replay, ICML, 2024

---

### Decision · Program_Chairs · 2024-09-25

**Decision:**

Accept (poster)

**Comment:**

This paper studies the dormant neuron phenomenon in value factorization of multi-agent reinforcement learning (MARL) and finds that the dormant neurons negatively affect the learning process. With observing that dormant neurons correlate with the existence of overweight neurons, the authors propose ReBorn that transfers the weights from overweight neurons to dormant neurons with some theoretical analysis and promising empirical results. Overall, the story path of this paper is interesting, and the proposed method is important for making CTDE MARL methods more effective. During the rebuttal phase, the authors provided satisfactory responses to ease the concerns of the reviewers. As a result, I recommend acceptance of this paper.